# Apelin inhibition prevents resistance and metastasis associated with anti-angiogenic therapy

Iris Uribesalgo[1,†,*], David Hoffmann[1,†], Yin Zhang[2,3], Anoop Kavirayani[4], Jelena Lazovic[5], Judit Berta[6], Maria Novatchkova[1], Tsung-Pin Pai[1], Reiner A Wimmer[1], Viktória László[7,8], Daniel Schramek[1,9], Rezaul Karim[1], Luigi Tortola[1], Sumit Deswal[10], Lisa Haas[10], Johannes Zuber[10], Miklós Szűcs[11], Keiji Kuba[1,12], Balazs Dome[6,7,13], Yihai Cao[2], Bernhard J Haubner[1,14] & Josef M Penninger[1,15,**]

## Abstract

Angiogenesis is a hallmark of cancer, promoting growth and metastasis. Anti-angiogenic treatment has limited efficacy due to therapy-induced blood vessel alterations, often followed by local hypoxia, tumor adaptation, progression, and metastasis. It is therefore paramount to overcome therapy-induced resistance. We show that Apelin inhibition potently remodels the tumor microenvironment, reducing angiogenesis, and effectively blunting tumor growth. Functionally, targeting Apelin improves vessel function and reduces polymorphonuclear myeloid-derived suppressor cell infiltration. Importantly, in mammary and lung cancer, Apelin prevents resistance to anti-angiogenic receptor tyrosine kinase (RTK) inhibitor therapy, reducing growth and angiogenesis in lung and breast cancer models without increased hypoxia in the tumor microenvironment. Apelin blockage also prevents RTK inhibitor-induced metastases, and high Apelin levels correlate with poor prognosis of anti-angiogenic therapy patients. These data identify a druggable anti-angiogenic drug target that reduces tumor blood vessel densities and normalizes the tumor vasculature to decrease metastases.

**Keywords** anti-angiogenic therapy; Apelin–Apelin receptor; therapy-induced resistance; tumor angiogenesis; VEGF-VEGFR
**Subject Categories** Cancer; Vascular Biology & Angiogenesis

See also: **L Claesson-Welsh** (August 2019)

## Introduction

Angiogenesis, the sprouting of new blood vessels from the existing vasculature, is a hallmark of cancer that facilitates rapid tumor growth and metastasis (Hanahan & Weinberg, 2011). Activation of an "angiogenic switch" during cancer progression causes aberrant capillary sprouting, tortuous and excessive vessel branching, enlarged vessels, erratic blood flow, micro-hemorrhages, leakiness, and abnormal endothelial cell proliferation (Hanahan & Weinberg, 2011). Inhibition of this angiogenic switch has therefore been proposed as a key cancer treatment strategy. Given the importance of vascular endothelial growth factors (VEGFs) in angiogenesis, much attention has been focused on developing anti-angiogenic receptor tyrosine kinase (RTK) inhibitors targeting the VEGFR signaling pathway to treat a variety of cancers by blocking tumor angiogenesis. Although various clinical trials have demonstrated the efficacy of these therapies, the benefits are usually transitory and, in certain cases, VEGFR pathway inhibitors may even lead to a more aggressive disease (Carmeliet & Jain, 2011a).

1   Institute of Molecular Biotechnology of the Austrian Academy of Sciences (IMBA), Vienna BioCenter, Vienna, Austria
2   Department of Microbiology, Tumor and Cell Biology, Biomedicum, Karolinska Institutet, Stockholm, Sweden
3   Medicine and Pharmacy Research Center, Binzhou Medical University, Yantai, Shandong Province, China
4   VBCF Histopathology, Vienna BioCenter, Vienna, Austria
5   VBCF Preclinical Imaging, Vienna BioCenter, Vienna, Austria
6   Department of Tumor Biology, National Koranyi Institute of Pulmonology, Budapest, Hungary
7   Division of Thoracic Surgery, Department of Surgery, Comprehensive Cancer Center, Medical University of Vienna, Vienna, Austria
8   Division of Molecular and Gender Imaging, Department of Biomedical Imaging and Image-guided Therapy, Medical University of Vienna, Vienna, Austria
9   Department of Molecular Genetics, Lunenfeld-Tanenbaum Research Institute, Mount Sinai Hospital, University of Toronto, Toronto, ON, Canada
10  Institute of Molecular Pathology (IMP), Vienna BioCenter, Vienna, Austria
11  Department of Urology, Semmelweis University, Budapest, Hungary
12  Department Biochemistry and Metabolic Science, Akita University Graduate School of Medicine, Akita, Japan
13  Department of Thoracic Surgery, National Institute of Oncology-Semmelweis University, Budapest, Hungary
14  Department of Internal Medicine III (Cardiology and Angiology), Medical University of Innsbruck, Innsbruck, Austria
15  Department of Medical Genetics, Life Science Institute, University of British Columbia, Vancouver, BC, Canada
    *Corresponding author. Tel: +43 (1)790 44; E-mail: iris.uribesalgo@imba.oeaw.ac.at
    **Corresponding author. Tel: +43 (1)790 44; E-mail: josef.penninger@imba.oeaw.ac.at
    †These authors contributed equally to this work

The large-scale eradication of tumor blood vessels results in necrosis and hypoxia, which can trigger several resistance mechanisms that drive tumor regrowth and malignancy (Bergers & Hanahan, 2008; Carmeliet & Jain, 2011a; Potente *et al*, 2011). In recent years, the concept of vessel normalization has been proposed to restore vascular abnormalities in tumors, with vessels becoming less permeable and better structured (Jain, 2001). Promoting vessel normalization has been linked to decreased metastasis and an increased efficacy of other therapies such as chemotherapies (Carmeliet & Jain, 2011b; Leite de Oliveira *et al*, 2012; Maes *et al*, 2014). However, treatment with VEGF/VEGFR inhibitors alone leads to a transient, usually rather short period of vessel normalization after which hypoxia recurs and acquired resistance emerges (Rivera & Bergers, 2015). In the last years, combination of VEGF and angiopoietin-2 (Ang2) blockage has shown greater effects than single targeting of both molecules in decreasing tumor growth, angiogenesis, vascular abnormality, and metastasis (Brown *et al*, 2010; Koh *et al*, 2010; Kienast *et al*, 2013; Rigamonti *et al*, 2014; Scholz *et al*, 2016; Allen *et al*, 2017; Schmittnaegel *et al*, 2017), thus increasing the promise of anti-angiogenic VEGF-targeting in combinatory therapies. However, this strategy increased hypoxia (Koh *et al*, 2010; Rigamonti *et al*, 2014; Scholz *et al*, 2016), which can worsen the tumor microenvironment and induce treatment resistance (Jain, 2014). Thus, there is a need to identify safe new agents that can retain the therapeutic advantages of current anti-angiogenic treatments, such as the reduction of angiogenesis and primary tumor growth, and at the same time prevent its resistance-associated features such as hypoxia and therapy-induced metastases.

Apelin is an evolutionarily conserved peptide that acts as the endogenous ligand for the G protein-coupled Apelin receptor (Tatemoto *et al*, 1998). The Apelin/Apelin receptor signaling pathway has been implicated in developmental angiogenesis (Saint-Geniez *et al*, 2002; Kasai *et al*, 2004, 2010; Cox *et al*, 2006; Kälin *et al*, 2007; Kidoya *et al*, 2008; del Toro *et al*, 2010; Kidoya & Takakura, 2012). Although the Apelin/Apelin receptor pathway is downregulated in adulthood, it is frequently reactivated and upregulated in tumors (Sorli *et al*, 2007; Berta *et al*, 2010), including in endothelial cells within the tumor microenvironment (Seaman *et al*, 2007). Further, elevated levels of Apelin are associated with poor clinical outcome in certain human cancers (Berta *et al*, 2010). Although these observations make the Apelin/Apelin receptor pathway a potentially attractive target for anti-angiogenic cancer therapy, the detailed effects of its targeting for cancer treatment *in vivo* are poorly understood. In addition, some reports suggest that the Apelin/Apelin receptor pathway is not redundant with VEGFR signaling and that both have independent roles in angiogenesis (Kidoya *et al*, 2008; del Toro *et al*, 2010; Heo *et al*, 2012). Therefore, we wanted to explore whether combinations of Apelin blockade with current anti-angiogenic therapies may be of therapeutic benefit in cancer.

Here, we show that genetic and pharmacological inhibition of Apelin is a feasible strategy to reduce tumor blood vessel formation, vessel leakiness, and hypoxia, as well as to reduce suppressive immune cell infiltration, thereby significantly diminishing growth of primary lung and mammary tumors. In 3D vascular sprouts, Apelin is essential for VEGF to trigger blood vessel outgrowth, indicating that Apelin might be a key pathway that interfaces with VEGF signaling. Combining targeting of Apelin with clinically relevant RTK inhibitors like sunitinib, *in vivo* not only reduced blood vessel density and leakage in tumors, but also decreased hypoxia and metastases induced by sunitinib treatment. Further, elevated Apelin levels in serum samples from renal cell cancer patients treated with sunitinib as a single agent were associated with a worse prognosis. Our findings unveil a new strategy that combines clinically relevant anti-angiogenic treatments with Apelin inhibition to diminish tumor growth, blood vessel density, and vessel abnormality within the tumor environment, and thus hypoxia, tumor resistance, and anti-angiogenic therapy-induced metastasis.

## Results

### Apelin blockage improves survival in mammary and lung cancer models

To corroborate that Apelin expression is associated with outcome in human breast cancer, we performed an unbiased meta-analysis of multiple datasets using the Kmplot (Györffy *et al*, 2010) and PrognoScan (Mizuno *et al*, 2009) databases. We confirmed that high levels of Apelin expression in tumors are significantly associated with poor prognosis in breast cancer patients (Fig EV1A). Next, we determined whether Apelin blockage is a suitable strategy to ameliorate cancer progression by ablating its expression in mammary cancer. Apelin-deficient ($Apln^{-/-}$) mice (Kuba *et al*, 2007) were crossed with MMTV-*NeuT* transgenic mice (Lucchini *et al*, 1992) to generate MMTV-*NeuT*; $Apln^{-/-}$ and MMTV-*NeuT*; $Apln^{+/+}$ control littermates (termed *NeuT;Apln*$^{-/-}$ and *NeuT;Apln*$^{+/+}$ hereafter). Apelin has been previously shown to be upregulated in tumor cells (Seaman *et al*, 2007; Wang *et al*, 2007; Liu *et al*, 2015). We confirmed that Apelin expression is enhanced in tumors of MMTV-*NeuT* mice compared to epithelial cells isolated from the mammary gland of healthy mice (Fig EV1B), recapitulating human breast cancer (Sorli *et al*, 2007) and validating our model. Importantly, *NeuT; Apln*$^{-/-}$ tumor-bearing mice displayed a delay in the onset of NeuT-driven mammary tumors and a significantly prolonged survival compared with *NeuT;Apln*$^{+/+}$ littermates (Figs EV1C and 1A). In line with enhanced survival, *Apelin*-null mice displayed a decreased tumor burden in the mammary glands compared to age-matched controls (Fig EV1D).

We further extended our studies to lung cancer as a second solid tumor model of epithelial origin. Similar to breast cancer, we confirmed that high levels of Apelin expression are significantly associated with poor prognosis in lung cancer patients (Fig EV1E) (Györffy *et al*, 2013). To be able to experimentally dissect the role of Apelin in lung cancer, $Apln^{-/-}$ mice were crossed to the *Lox-Stop-Lox-KRas*$^{G12D}$ lung cancer model (*KRas;Apln*$^{+/y}$ and *KRas;Apln*$^{-/y}$ hereafter; the Apelin gene is located on the X chromosome; Kuba *et al*, 2007) (Jackson, 2001). We also extended the investigation to a more aggressive form of non-small cell lung cancer (NSCLC) driven by the KRas$^{G12D}$ oncogene combined with loss of the tumor suppressor p53 (*p53*$^{f/f}$;*KRas;Apln*$^{+/y}$ and *p53*$^{f/f}$;*KRas;Apln*$^{-/y}$; DuPage *et al*, 2009). Knockout of Apelin resulted in enhanced survival and reduced tumor burden in these lung cancer models (Fig EV1F–H).

We also explored whether *Apela*, the recently described second ligand for Apelin receptor (Pauli *et al*, 2014), might also be overexpressed in NeuT-driven mammary tumors or KRas-driven lung tumors, but we failed to detect its expression, even using sensitive

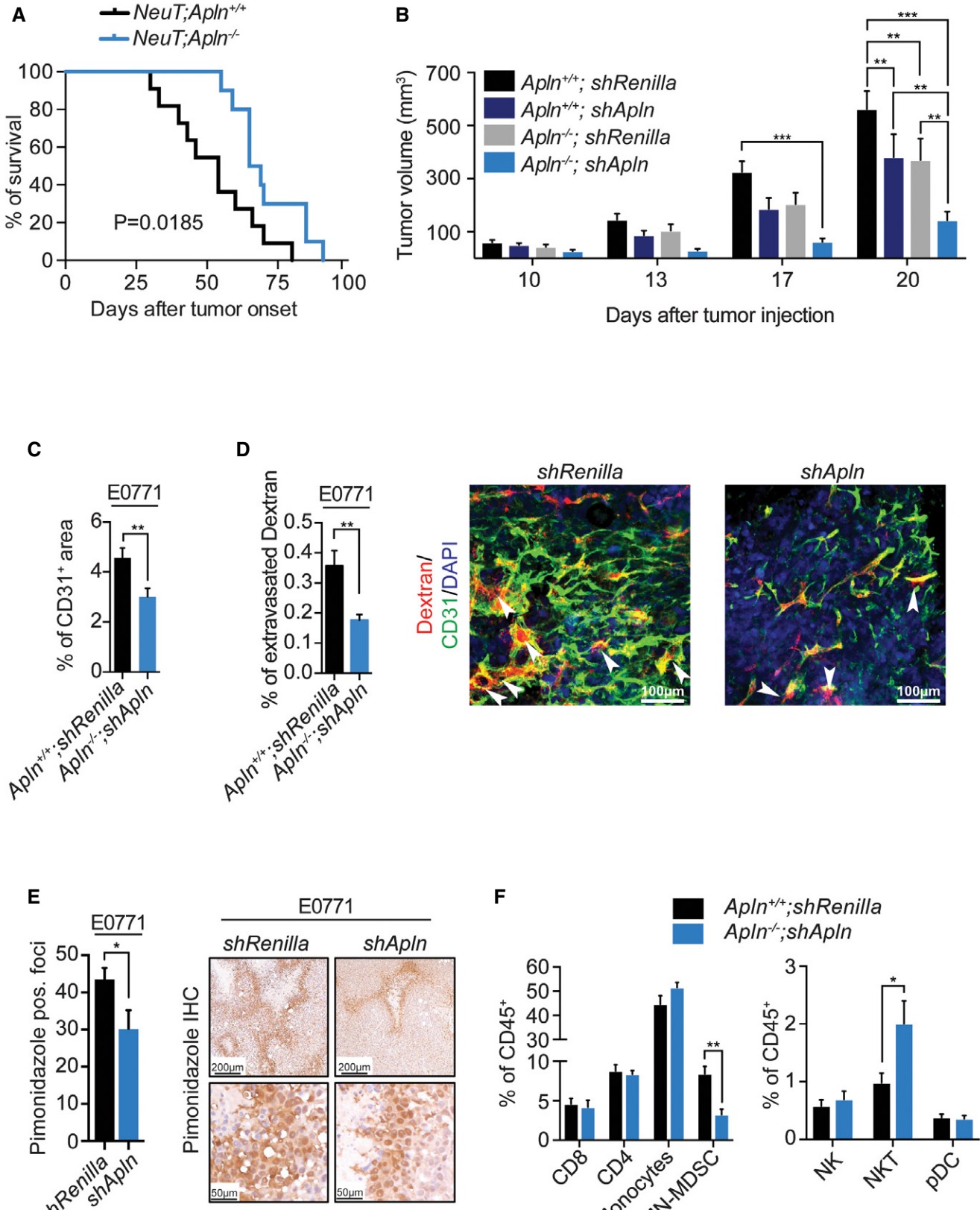

Figure 1.

◀

Figure 1. **Genetic and pharmacological inhibition of Apelin impairs mammary tumor growth and tumor angiogenesis in a paracrine manner.**

A   Kaplan–Meier plot for survival in $NeuT;Apln^{+/+}$ ($n = 11$) and $NeuT;Apln^{-/-}$ ($n = 10$) mice with mammary cancer after tumor onset. *$P = 0.0185$; log-rank test. Mice were sacrificed when the tumor size reached 1 cm$^3$, following ethical guidelines.

B   Tumor volumes, followed over time, of control mammary tumor E0771 cells ($shRenilla$) and Apelin-depleted ($shApln$) E0771 cells orthotopically injected into both syngeneic C57BL/6J $Apln^{+/+}$ and $Apln^{-/-}$ mice ($5 \times 10^5$ cells/mouse), respectively. Tumor volumes were determined using calipers and are shown as mean tumor volumes ± SEM. Data shown are pooled from two independent experiments. $Apln^{+/+};shRenilla$ ($n = 18$), $Apln^{+/+};shApln$ ($n = 17$), $Apln^{-/-};shRenilla$ ($n = 14$), $Apln^{-/-};shApln$ ($n = 15$); **$P < 0.01$, ***$P < 0.001$; two-way ANOVA.

C   Mean percentages (±SEM) of CD31$^+$ area in E0771 $shRenilla$ ($n = 3$) or $shApln$ ($n = 3$) mammary tumors, assessed on day 23 post-orthotopic injection into C57BL/6J $Apln^{+/+}$ or $Apln^{-/-}$ mice, respectively. **$P < 0.01$; t-test.

D   Mean percentages (±SEM) of extravasated Dextran in E0771 $shRenilla$ ($n = 9$) or $shApln$ ($n = 12$) mammary tumors, assessed on day 19 post-orthotopic injection into C57BL/6J $Apln^{+/+}$ or $Apln^{-/-}$ mice, respectively. **$P < 0.01$; t-test. Right panel shows representative immunofluorescence of Dextran (red), CD31$^+$ vessels (green), and DAPI (blue). The white arrows indicate regions of Dextran extravasation. Scale bars = 100 µm.

E   Mean counts (±SEM) of pimonidazole positive foci, assessed on day 26 post-orthotopic injection of E0771 $shRenilla$ ($n = 6$) or $shApln$ ($n = 4$) into C57BL/6J wild-type mice ($5 \times 10^5$ cells/mouse). *$P < 0.05$; t-test. Right panels show representative immunohistochemical pimonidazole staining at two different magnifications; scale bars = 200 µm (upper panels) and 50 µm (lower panels).

F   Mean percentage (±SEM) of tumor-infiltrating immune cells normalized to CD45$^+$. E0771 $shRenilla$ ($n = 8$) or $shApln$ ($n = 6$) were orthotopically injected into C57BL/6J $Apln^{+/+}$ or $Apln^{-/-}$ mice ($5 \times 10^5$ cells/mouse), respectively, and tumors were harvested day 25 post-injection. *$P < 0.05$, **$P < 0.01$; t-test. All immune cell populations were gated for viable CD45$^+$ cells and then further defined as: CD8 T cells (Thy1.2$^+$, CD8$^+$), CD4 T cells (Thy1.2$^+$, CD4$^+$), inflammatory monocytes (Ly6C$^+$, Cd11b$^+$, Ly6G$^-$), PMN-MDSCs (Ly6G$^+$, Cd11b$^+$), natural killer cells (Thy1.2$^-$, Ly6G$^-$, NK1.1$^+$), natural killer T cells (Thy1.2$^+$, CD4$^-$, CD8$^-$, NK1.1$^+$), and peripheral dendritic cells (Ly6G$^-$,Ly6C$^+$, Cd11b$^-$, PDAC1$^+$, B220$^+$).

Source data are available online for this figure.

and multi-cycle qPCR analysis. Thus, we conclude that Apelin is the primary Apelin receptor ligand upregulated in our models of mammary and lung cancer. Of note, we did not detect abnormalities in mammary glands or lungs from non-tumor-bearing adult and background-matched $Apln^{+/+}$ and $Apln^{-/-}$ mice without oncogenic drivers. These results not only extend the findings of previous over-expression studies (Sorli *et al*, 2007; Berta *et al*, 2010), but validate Apelin as a target in tumor models of epithelial origin. Thus, high Apelin levels correlate with a worse prognosis in breast and lung cancer patients, and Apelin inactivation increases the survival of mice with breast and lung cancer.

## Apelin modulates the tumor microenvironment through paracrine stimulation of tumor angiogenesis

Apelin has previously been shown to stimulate tumor angiogenesis and is upregulated in tumor-associated endothelial cells (Seaman *et al*, 2007; Wang *et al*, 2007; Liu *et al*, 2015), which we could confirm in endothelial cells isolated from MMTV-NeuT tumors compared to normal mammary gland (Fig EV2A). Despite the known role of Apelin in tumor angiogenesis, the detailed effects of Apelin within the tumor cells and its microenvironment *in vivo* remain poorly understood.

To study the contribution to tumor growth of Apelin from cancer cells or cells of the tumor microenvironment, we used the E0771 mammary cancer model and, first, specifically downregulated the expression of *Apln* in cancer cells using shRNA (Fig EV2B). Then, we orthotopically injected control *shRenilla* E0771 cells and *Apln*-depleted (*shApln*) E0771 cells into syngeneic C57BL/6J $Apln^{+/+}$ as well as $Apln^{-/-}$ mice (Fig 1B). Comparing Apelin wild-type ($Apln^{+/+}$;E0771 *shRen*) with Apelin-depleted ($Apln^{-/-}$;E0771 *shApln*) tumors, we could recapitulate the phenotype observed with the constitutive Apelin knockout in the *NeuT* model (Fig 1A and B). By specifically depleting Apelin expression in tumor epithelial cells ($Apln^{+/+}$;E0771 *shApln*) or the cells of the tumor microenvironment ($Apln^{-/-}$;E0771 *shRen*), we could investigate the importance of these two sources of Apelin for the observed growth decrease in fully Apelin-depleted tumors ($Apln^{-/-}$;E0771 *shApln*). We found

that both Apelin sources are of equal importance for the Apelin-mediated increase in tumor growth (Fig 1B).

While the Apelin receptor has been detected in both tumor and endothelial cells, we find that its expression was considerably higher in tumor-associated endothelial cells (Fig EV2C). To assess whether Apelin exerts its effects in the tumor epithelial cells or in the tumor endothelial cells, we used the E0771 mammary cancer model and specifically downregulated the expression of the Apelin receptor (*Aplnr*) in the cancer cells using shRNA (Fig EV2B). Whereas *in vitro* shRenilla, *shApln* and *shAplnr* E0771 cells grew similarly (Fig EV2D), tumors from injected *shAplnr* E0771 cells in syngeneic wild-type mice did not show a reduction in tumor growth compared to tumors from injected *shRenilla* E0771 cells, in contrast to tumors from *shApln* E0771 cells (Fig EV2E). In addition, only tumors from *shApln* E0771 cells presented a decreased microvessel density (Fig EV2F), indicating that tumor epithelial cell-derived Apelin induces tumor angiogenesis in a paracrine fashion.

Importantly, loss of Apelin expression also significantly decreased microvessel densities in both E0771 and NeuT-driven mammary tumors, as well as KRas$^{G12D}$-driven lung tumors (Fig 1C, and Appendix Fig S1A and B). Functionally, E0771 *shApln* cells injected into $Apln^{-/-}$ mice displayed markedly reduced vessel leakage *in vivo* as compared to control *shRenilla* E0771 cells injected into $Apln^{+/+}$ mice (Fig 1D). Using pimonidazole staining to detect tissue hypoxia (Varia *et al*, 1998) revealed a decreased number of hypoxic foci in *shApln* E0771 mammary tumors (Fig 1E).

Angiogenic proteins, like VEGF, have been reported to be able to affect immune cell infiltration in different tumor models (Yang *et al*, 2018). To complement our findings, we profiled the immune cell infiltration in the $Apln^{+/+}$;E0771 *shRenilla* and $Apln^{-/-}$;E0771 *shApln* tumor groups. While total immune cell infiltration, as determined by the numbers of CD45$^+$ cells in the tumor, was unchanged (Appendix Fig S1C), we found a significant decrease of polymorphonuclear myeloid-derived suppressor cells (PMN-MDSC) and a significant increase in NK T cells in tumor from Apelin-depleted mice (Fig 1F). Of note, it has been previously reported that PMN-MDSC cells accumulate in hypoxic tumor regions and are associated

with increased angiogenesis *in vivo* and enhanced tumor cell invasion (Marvel & Gabrilovich, 2015). Together, these results show that tumor cell-derived as well as microenvironment-derived Apelin contributes to cancer progression through stimulation of tumor angiogenesis, enhancing vessel leakiness and tumor hypoxia, and altered infiltration of immune cells.

## Apelin induces pro-angiogenic pathways in endothelial cells and enhances VEGF-induced vessel sprouting

Having established that Apelin is a modulator of tumor blood vessels, we next explored gene expression changes of $CD31^+/CD105^+$ endothelial cells (ECs) sorted from Apelin wild-type and Apln-depleted tumors. We used ingenuity pathway analysis (IPA) to predict regulation of downstream biological processes and found a significant decrease in processes associated with endothelial cell proliferation and angiogenesis in ECs sorted out of Apelin-depleted tumors (Fig 2A), consistent with our previous findings (Fig 1C, Appendix Fig S1A and B). Further, IPA predicted a decrease in the adhesion of granulocytes (the cellular family to which PMN-MDSCs belong), also in line with our findings (Figs 1F and 2A). IPA is also suitable to predict upstream regulators and their activation state based on the up- or downregulation of differentially expressed genes. We find that proteins of the TGF superfamily, Inhibin-βA and TGF-β1, as well as C/EBP-α, β-catenin, ErbB2 and EGFR are predicted to be inhibited upstream regulators in ECs isolated from Apelin-depleted tumors (Fig EV3A). Thus, Apelin depletion results in impaired angiogenesis as determined by decreased blood vessel numbers and transcriptome analysis of tumor-associated endothelial cells.

To further investigate the role of Apelin in angiogenesis, we used an *in vitro* system of 3D vessel sprouting from embryoid bodies (EBs) that allowed us to study active angiogenesis in a controlled environment, mimicking *in vivo* vessel growth (Jakobsson *et al*, 2010). Although Apelin stimulation alone was not sufficient to initiate vessel sprouting, we found that it strongly increased VEGF-dependent vessel growth (Fig EV3B). We next generated Apelin mutant mouse embryonic stem cells (mESCs) using a gene-trap strategy in haploid stem cells (Elling *et al*, 2011, 2017). A sense splice acceptor disrupted Apelin mRNA expression (*Apln* STOP mESCs), whereas Cre expression "genetically repaired" Apelin expression in sister cells (*Apln* GO mESCs; Fig EV3C and D). Although *Apln* STOP mESCs displayed comparable growth kinetics (Fig EV3E), *Apln* STOP EBs exhibited markedly delayed sprouting upon VEGF treatment compared with their *Apln* GO controls (Fig 2B). Of note, Cre expression *per se* did not affect vessel sprouting compared to a non-Cre control. Thus, Apelin enhances VEGF-induced vessel sprouting *in vitro*.

To characterize the functional behavior of endothelial cells with or without Apelin, we performed competition assays in which chimeric EBs were established by mixing *Apln* GO: *Apln* STOP sister mESCs 1:1 followed by stimulation with VEGF. To track *Apln* GO and *Apln* STOP cells, we incorporated a GFP signal in our gene-trap strategy and a mCherry signal in our retroviral Cre vector; *Apln* STOP targeted cells are $GFP^+$ and *Apln* GO cells are $mCherry^+$ (Fig EV3C). Chimeric EBs treated with VEGF showed a significant decrease in the ratio of *Apln* STOP ($GFP^+$):*Apln* GO ($mCherry^+Cre^+$) endothelial cells by FACS analysis (Fig EV3F), confirming

the functional disadvantage of Apelin STOP cells in vessel sprouting.

In line with *in vivo* EC RNA sequencing (RNA-Seq) and delayed vessel sprouting, downregulated genes of purified $CD31^+$ ECs isolated from sprouting *Apln* STOP EBs treated with VEGF showed angiogenesis as the top affected gene ontology category compared with *Apln* GO control cells (Fig 2C). In addition, loss of Apelin modulated pathways in endothelial cells related to vasculogenesis, cell adhesion, and response to hypoxia (Fig 2C). IPA identified VEGF as the main upstream regulator predicted to be inhibited in Apelin-depleted endothelial cells, closely followed by TGFβ1 and TNF, all reduced in the absence of Apelin (Fig EV3G). Indeed, genes that IPA predicted to be downstream of VEGF showed strong downregulation in Apelin-depleted ECs and were enriched for angiogenesis-related genes (Fig EV3H). Thus, in a controlled model of VEGF-induced angiogenesis, Apelin and VEGF induce similar downstream pathways and genes relevant for endothelial cell proliferation and blood vessel sprouting, suggesting that Apelin inhibition could complement and potentiate current anti-angiogenic cancer treatment.

## Apelin ablation enhances effectiveness of anti-angiogenic treatment

Since we observed that Apelin depletion decreased VEGF-dependent vessel sprouting (Figs EV3C–H, and 2B and C), we aimed to study the effects of targeting Apelin in tumors in combination with clinically used anti-angiogenic treatments using inhibitors targeting receptor tyrosine kinases (RTKIs). We selected sunitinib, an inhibitor of multiple receptor tyrosine kinases implicated in angiogenesis, including VEGFR1, VEGFR2, and VEGFR3, PDGFRα, PDGFRβ, Kit, and others (Kim *et al*, 2014). Further, sunitinib is approved for clinical use in renal cell carcinoma, gastrointestinal stromal tumor and pancreatic neuroendocrine tumors (Rock *et al*, 2007; de Wilde *et al*, 2012).

To determine if blocking the Apelin pathway could complement current anti-angiogenic therapies, we administered sunitinib to $NeuT;Apln^{-/-}$ and $NeuT;Apln^{+/+}$ mice at the time of mammary tumor onset. In $NeuT^+$ mice, Apelin ablation combined with sunitinib treatment significantly increased survival and reduced the tumor burden compared with either intervention alone (Fig 3A and B); blocking Apelin almost doubled the survival of sunitinib-treated mice and tripled the survival compared to control untreated mice (Fig 3A). It should be noted that mice had to be sacrificed when reaching a particular tumor size following ethical guidelines. Similar results were observed with E0771 mammary cancer cells (Fig EV4A) and with axitinib, a second clinically used anti-angiogenic treatment (Fig EV4B), extending our findings to several cancer models and drugs. To further test whether Apelin cooperates with VEGFR signaling and whether the combinational blockage of both could be exploited in anti-cancer treatment, we treated *shRenilla* and *shApln* E0771 cell-bearing C57BL/6J mice with a specific anti-VEGFR2 antibody (DC101 clone) and obtained similar results as with sunitinib and axitinib treatments (Fig EV4C). These data show that combining Apelin inhibition with other inhibitors of angiogenesis markedly delays mammary cancer growth.

We next followed individual tumors *in vivo* using non-invasive magnetic resonance imaging (MRI). MRI analysis of tumor volume

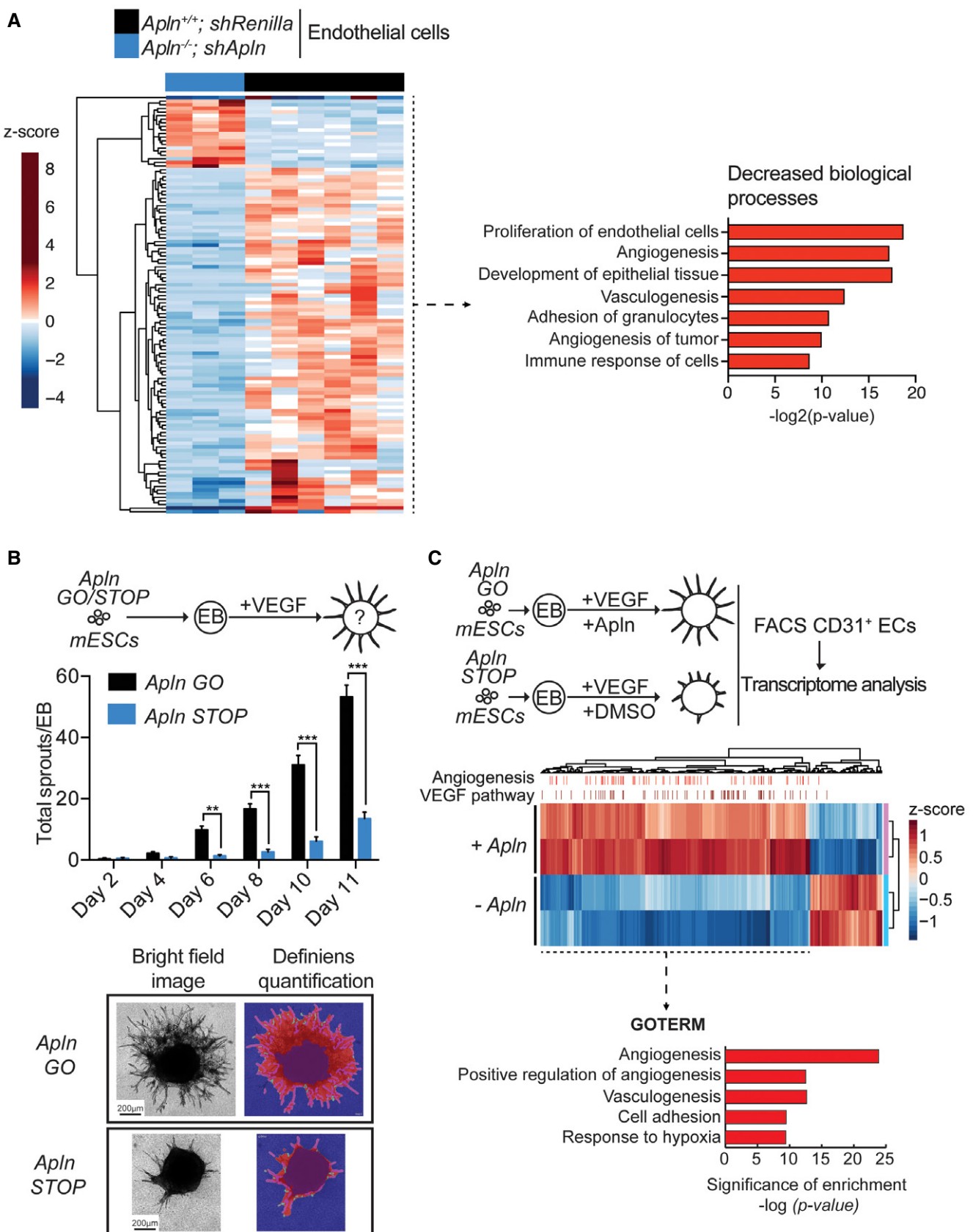

Figure 2.

**Figure 2. Apelin deletion delays VEGF-induced blood vessel sprouting.**

A  Left—Heatmap of RNA-Seq transcriptome analysis of CD31$^+$/CD105$^+$ endothelial cells sorted from tumors established by E0771 *shRenilla* ($n = 6$) or *shApln* ($n = 3$) cells orthotopically injected into C57BL/6J *Apln$^{+/+}$* or *Apln$^{-/-}$* mice, respectively. Tumors were harvested day 25 post-injection and genes displayed are significantly deregulated at the adjusted *P*-value cutoff of 0.05. Right—ingenuity pathway analysis for biological processes predicted to be decreased downstream of the differentially expressed genes.

B  Quantification of vessel sprouts (mean values ± SEM) upon VEGF treatment (30 ng/ml) of embryoid bodies (EB) derived from murine ES cells (mESCs) with sense integrations in the Apelin gene (*Apln* STOP) of the splice acceptor described in Appendix Fig S1B or Cre-reverted antisense (*Apln* GO) sister cells. *Apln* GO (Day 2 $n = 6$; Day 4 $n = 40$, Day 6 $n = 54$, Day 8 $n = 32$, Day 10 $n = 35$, Day 11 $n = 31$), *Apln* STOP (Day 2 $n = 4$; Day 4 $n = 15$, Day 6 $n = 31$, Day 8 $n = 28$, Day 10 $n = 32$, Day 11 $n = 37$); **$P < 0.01$, ***$P < 0.001$, two-way ANOVA. Representative brightfield images and automated analysis of vessel sprouts by Definiens software are shown in the bottom panels. Scale bars = 200 μm.

C  Differentially expressed genes using RNA-Seq transcriptome analysis of CD31$^+$ endothelial cells (ECs) sorted from sprouting EBs from Apelin STOP cells stimulated with VEGF (30 ng/ml) and DMSO (−Apln) and repaired Apelin GO sister cells stimulated with VEGF and Apelin (1,000 nM; +Apln). VEGF target genes and angiogenesis-related genes, predicted by ingenuity pathway analysis (IPA) software, are indicated by bars on the upper axis of the heatmap. GO terms were analyzed by DAVID online software.

confirmed that inactivation of Apelin decreased the growth rate of NeuT$^+$ mammary tumors to levels comparable to sunitinib treatment (Fig 3C and Movie EV1). Importantly, sunitinib-treated *NeuT; Apln$^{-/-}$* tumors showed a significant decrease in tumor growth (Fig 3C and Movie EV1), as well as reduced mitotic counts and Ki67 staining (Figs 3D and EV4D). Decreased mitotic counts and Ki67 levels are established markers of better prognosis in breast cancer (van Diest *et al*, 2004). Finally, we used MM54, an Apelin antagonist with a K$_D$ of 3.4 μM and no reported agonistic activity, to block Apelin signaling (Macaluso *et al*, 2011; Harford-Wright *et al*, 2017). Pharmacologic inhibition of Apelin signaling by MM54 had the same effect as genetically ablating Apelin expression, synergizing with sunitinib to reduce tumor progression (Fig 3E). We obtained similar results on overall survival, tumor growth, and tumor cell proliferation in the *p53$^{f/f}$;KRas* lung cancer model (Appendix Fig S2A–C). These data in mammary cancer and KRas-driven lung cancer models indicate that inhibition of Apelin signaling improves the efficacy of anti-angiogenic therapy to impair primary tumor growth and promote survival.

**Combined Apelin and VEGFR inhibition normalizes blood vessels and prevents hypoxia in tumors**

Previous findings have shown that RTKIs, despite reducing tumor vasculature and growth, have no or limited beneficial effects in cancer treatment. As one explanation, it has been reported that anti-VEGFR treatment results in local hypoxia and consequently more metastases (Bergers & Hanahan, 2008). Therefore, we examined the tumor microenvironment and tumor cell dissemination. Having shown that inhibition of Apelin and administration of sunitinib markedly improves the outcome in different cancer models, we first assessed the impact of this combination therapy on the tumor vasculature. Size-matched early stage tumors were imaged by 15.2 Tesla MRI at 2, 4, and 6 weeks, and the relative tumor blood volume (rTBV) was assessed (Fig 4A). Whereas control *NeuT; Apln$^{+/+}$* tumors showed an increase in rTBV over time, sunitinib treatment reduced the rTBV in the tumors (Fig 4B and C), confirming previous data (Pàez-Ribes *et al*, 2009). *NeuT;Apln$^{-/-}$* tumors also displayed reduced rTBV, though to a lesser extent. Notably, sunitinib treatment of *NeuT;Apln$^{-/-}$* tumors enhanced the decrease in rTBV compared to either condition alone, and this was maintained over the entire observation period (Fig 4B and C). We corroborated the rTBV MRI analysis with anti-CD31 immunostaining

to determine blood vessel density within tumors (Fig 4D and E). In contrast to sunitinib-treated *NeuT;Apln$^{+/+}$* tumors that displayed low vessel density and poor vessel structure, characterized by dilated blood vessels and reduced mural cell coverage, as determined by the ratio of αSMA$^+$-positive vascular smooth muscle cells to CD31$^+$ endothelial cells, sunitinib-treated *NeuT;Apln$^{-/-}$* tumors exhibited markedly lower vessel densities and, importantly, normalized blood vessel structures (Fig 4D and E).

Next, we examined the functional consequences of combined Apelin inactivation and anti-angiogenic therapy on local hypoxia and leakage of the tumor vasculature. Sunitinib treatment of *NeuT; Apln$^{+/+}$* tumors increased the percentage of hypoxic cells adjacent to tumor blood vessels compared to untreated *NeuT;Apln$^{+/+}$* tumors, evaluated by prototypic carbonic anhydrase 9 (CA9) immunostaining (Fig 5A; Olive *et al*, 2001). In contrast, loss of Apelin reduced the percentage of hypoxic cells adjacent to tumor blood vessels in untreated and, most substantially, in sunitinib-treated *NeuT;Apln$^{-/-}$* tumors (Fig 5A). To complement these studies, we used MRI to visualize the dynamics of vessel permeability over time upon sunitinib therapy and Apelin targeting, following the same scheme used for rTBV assessment (see Fig 4A). Systematic dynamic susceptibility contrast (DSC) perfusion analysis in tumor tissue by MRI confirmed that sunitinib-treated *NeuT;Apln$^{-/-}$* tumors have the lowest vessel permeability (Fig 5B and C). Together, our data reveal that combined inhibition of angiogenic pathways not only reduces tumor growth and angiogenesis but also improves blood vessel structure relative to either condition alone. Importantly, loss of Apelin prevents the detrimental effects of sunitinib treatment on blood vessel structure, leakage, and local hypoxia.

**Apelin loss prevents sunitinib-induced metastases**

Tumors treated with sunitinib display increased hypoxia and invasiveness upon therapy-induced resistance, resulting in more metastasis, which cannot be reverted by stopping the treatment (Ebos *et al*, 2009; Pàez-Ribes *et al*, 2009). Therefore, we next evaluated how sunitinib treatment and loss of Apelin signaling affect the metastatic status of *NeuT;Apln$^{+/+}$* and *NeuT;Apln$^{-/-}$* mice. We analyzed lungs of tumor-bearing mice 6 weeks after the primary mammary tumors were size-matched at 20–70 mm$^3$ (see Fig 6A). Although untreated *NeuT;Apln$^{+/+}$* mammary tumors are not highly metastatic, sunitinib treatment significantly increased the number of

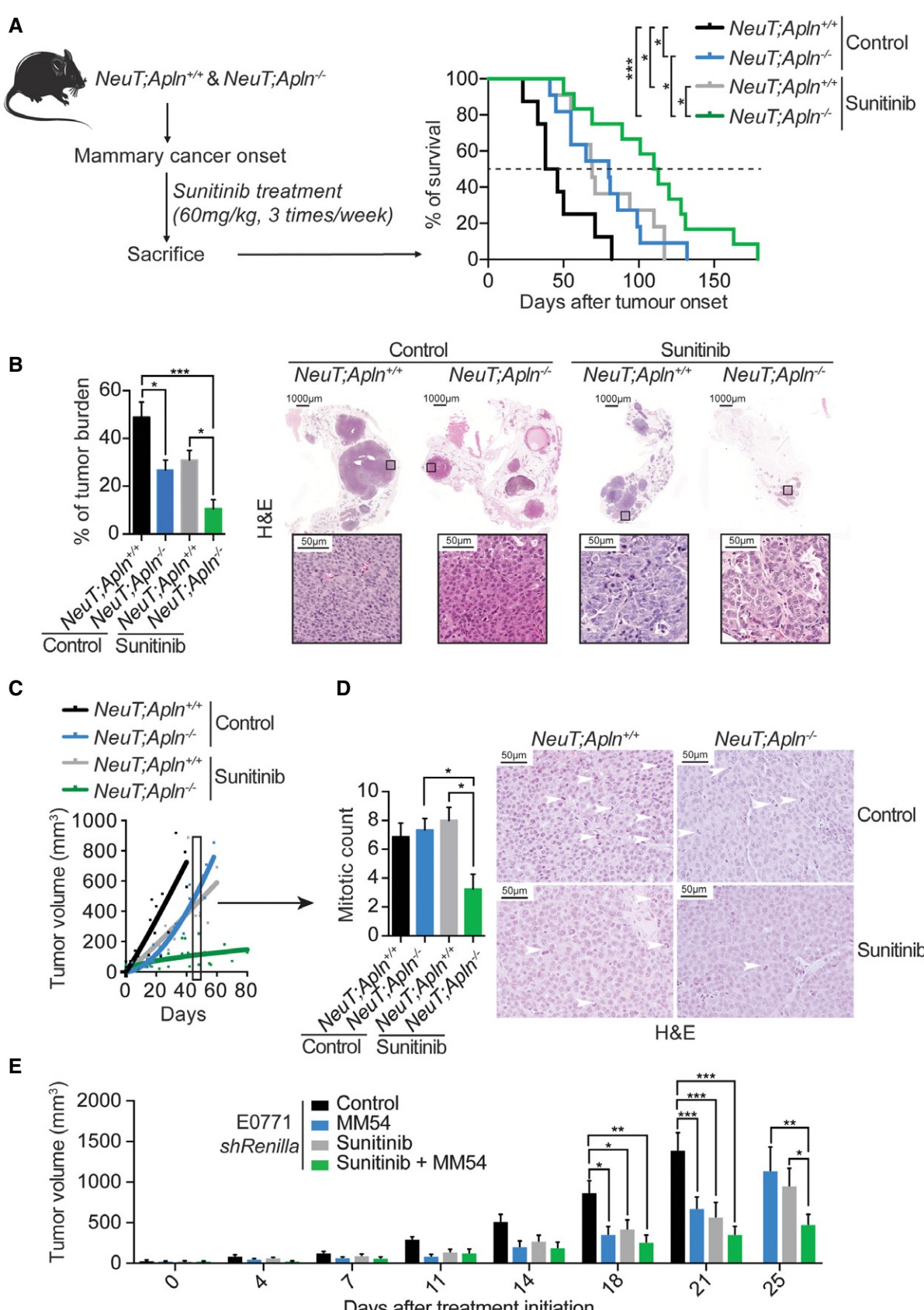

**Figure 3.**

**Figure 3. Combining Apelin blockage and anti-angiogenic treatment improves survival and reduces mammary tumor growth.**

A  Experimental setup and (right) Kaplan–Meier survival plot of *NeuT;Apln$^{+/+}$* and *NeuT;Apln$^{-/-}$* mice with mammary cancer, left untreated (control) or treated with the indicate dose of the broad VEGFR blocker sunitinib. *NeuT;Apln$^{+/+}$* control (*n* = 8), *NeuT;Apln$^{-/-}$* control (*n* = 11), *NeuT;Apln$^{+/+}$* sunitinib (*n* = 11), *NeuT;Apln$^{-/-}$* sunitinib (*n* = 12); *$P$ < 0.05; ***$P$ < 0.001; log-rank test. Mice were sacrificed when the tumor size reached 1 cm³, following ethical guidelines. The dotted line indicates 50% of survival.

B  Percentages (mean ± SEM) of tumor burden in mammary glands of untreated (control) and sunitinib-treated *NeuT;Apln$^{+/+}$* and *NeuT;Apln$^{-/-}$* mice, assessed 4 weeks after tumor onset. *NeuT;Apln$^{+/+}$* control (*n* = 6), *NeuT;Apln$^{-/-}$* control (*n* = 8), *NeuT;Apln$^{+/+}$* sunitinib (*n* = 5), *NeuT;Apln$^{-/-}$* sunitinib (*n* = 5); *$P$ < 0.05; ***$P$ < 0.001; one-way ANOVA. Representative H&E images are shown for each genotype in the right panels. Scale bars = 1,000 μm (large panels) and 50 μm (insets).

C  Tumor volumes of untreated (control) and sunitinib-treated *NeuT;Apln$^{+/+}$* and *NeuT;Apln$^{-/-}$* mammary tumors, size-matched at 20–70 mm³ and followed over time by MRI analysis; *NeuT;Apln$^{+/+}$* control (*n* = 7), *NeuT;Apln$^{-/-}$* control (*n* = 5), *NeuT;Apln$^{+/+}$* sunitinib (*n* = 5), *NeuT;Apln$^{-/-}$* sunitinib (*n* = 6) mice per group; lines represent nonlinear fit of tumor growth. Box and arrow indicate the time point used for analysis in panel (D).

D  Mitotic counts (mean ± SEM) and representative H&E images of mammary tumors in untreated (control) and sunitinib-treated *NeuT;Apln$^{+/+}$* and *NeuT;Apln$^{-/-}$* mice, assessed 6 weeks after tumor onset.; *NeuT;Apln$^{+/+}$* control (*n* = 7), *NeuT;Apln$^{-/-}$* control (*n* = 6), *NeuT;Apln$^{+/+}$* sunitinib (*n* = 9), *NeuT;Apln$^{-/-}$* sunitinib (*n* = 4) tumors per group; *$P$ < 0.05; one-way ANOVA to sunitinib-treated *NeuT;Apln$^{-/-}$*. White arrows indicate mitotic figures. Scale bars = 50 μm.

E  Tumor volumes, followed over the indicated time, of orthotopically injected E0771 *shRenilla* cells left untreated (control) or treated three times a week from day 8 after tumor injection with an Apelin antagonist alone (MM54, 0.4 μg/g), sunitinib alone (60 mg/kg) or a combination of both. Tumor volumes were measured using calipers and are shown as mean tumor volumes ± SEM. E0771 *shRenilla* control (*n* = 6), MM54 (*n* = 5), sunitinib (*n* = 5), sunitinib + MM54 (*n* = 5); *$P$ < 0.05, **$P$ < 0.01, ***$P$ < 0.001, two-way ANOVA.

Source data are available online for this figure.

lung metastases (Fig 6A), as previously observed (Ebos *et al*, 2009; Pàez-Ribes *et al*, 2009). Apelin inactivation alone did not significantly alter metastasis of NeuT-driven tumors. However, deletion of Apelin prevented sunitinib-induced lung metastases (Fig 6A). We obtained similar results in *NeuT$^+$* mice with large size-matched mammary tumors (Fig EV5A). Further, we found that sunitinib treatment induced metastasis of tumors derived from *shRenilla* E0771 cells but not from *shApln* E0771 cells, providing additional evidence that Apelin inhibition prevents sunitinib-induced metastasis (Fig EV5B and C).

To complement our studies in mice, we investigated whether Apelin levels correlate with the metastatic status of women with breast cancer. Unbiased meta-analysis using the Kmplot database (Győrffy *et al*, 2010) indeed showed that high levels of intratumoral Apelin expression significantly associated with an accelerated appearance of distant metastases in breast cancer (Fig 6B). These results suggest that Apelin levels are a potential prognostic biomarker for metastasis, with higher levels correlating with a shorter time to metastasis in breast cancer patients. Our experimental mouse results show that, unlike sunitinib treatment, loss of Apelin does not increase metastasis. Rather, Apelin inhibition not only reduces tumor growth in our mammary tumor models, but also reduces the occurrence of anti-angiogenic therapy-induced metastases.

**High Apelin levels correlate with poor prognosis of patients on sunitinib therapy**

Sunitinib is a first-line therapy for renal cell carcinoma (RCC) as a single agent (Kim *et al*, 2014). However, even in the treatment of RCC patients, it only temporarily stabilizes the disease (Kim *et al*, 2014). Based on our results, we hypothesized that low Apelin levels may correlate with a better prognosis in patients treated with anti-angiogenic therapy. To test this hypothesis, we measured Apelin levels in serum samples from a cohort of 55 RCC patients that were treated for 3–5 months with sunitinib as a single agent (Appendix Fig S3A) and evaluated their progression-free survival (PFS). RCC patients with lower Apelin levels indeed had a significantly longer PFS (median survival = 459.5 days) than patients with

higher Apelin levels (median survival = 280 days; Fig 6C and Appendix Fig S3B). We also evaluated the serum levels of both VEGF and Apelin in our cohort of sunitinib-treated RCC patients and stratified high or low expression groups based on the median (Fig 6D). Patients with both low Apelin and low VEGF serum levels had a significantly higher PFS (median survival = 623 days), whereas patients with high serum levels of both proteins had the lowest median survival (167 days). Low levels of only Apelin or VEGF showed a median survival of 340.5 and 343 days, respectively (Fig 6D). Taken together, these results indicate that high Apelin levels in serum samples correlate with worse prognosis of renal cancer patients treated with approved and clinically utilized anti-angiogenic therapy.

## Discussion

Tumor angiogenesis is required to nourish rapidly growing transformed cells. Anti-angiogenic therapies currently in clinical use are based on the inhibition of VEGF and closely related pathways (Cao *et al*, 2011; Jayson *et al*, 2016). However, in many cases, anti-angiogenic therapies result in abnormal blood vessels and local hypoxia in tumors, leading to even more aggressive growth and consequently metastases, thereby limiting the promise of blood vessel targeted anti-cancer approaches (Bergers & Hanahan, 2008; Carmeliet & Jain, 2011a; Potente *et al*, 2011; Rivera & Bergers, 2015). Emerging evidence shows that combinational therapies have usually higher effectiveness than monotherapies for cancer treatment. Based on this concept, we aimed to identify novel druggable pathways that can alleviate the detrimental effects of current anti-angiogenic therapies while maintaining their efficacy. Here, we show that ablating the small peptide Apelin strikes one Achilles heel of current anti-angiogenic therapy, reducing tumor angiogenesis and growth of the primary tumor while at the same time maintaining a better structured vasculature with higher pericyte coverage, impairing malignant progression, preventing hypoxia and distant metastasis.

One intriguing aspect of our findings is that inactivation of Apelin not only reduces tumor growth and cancer neo-angiogenesis in combination with anti-angiogenic therapy, but results in tumor

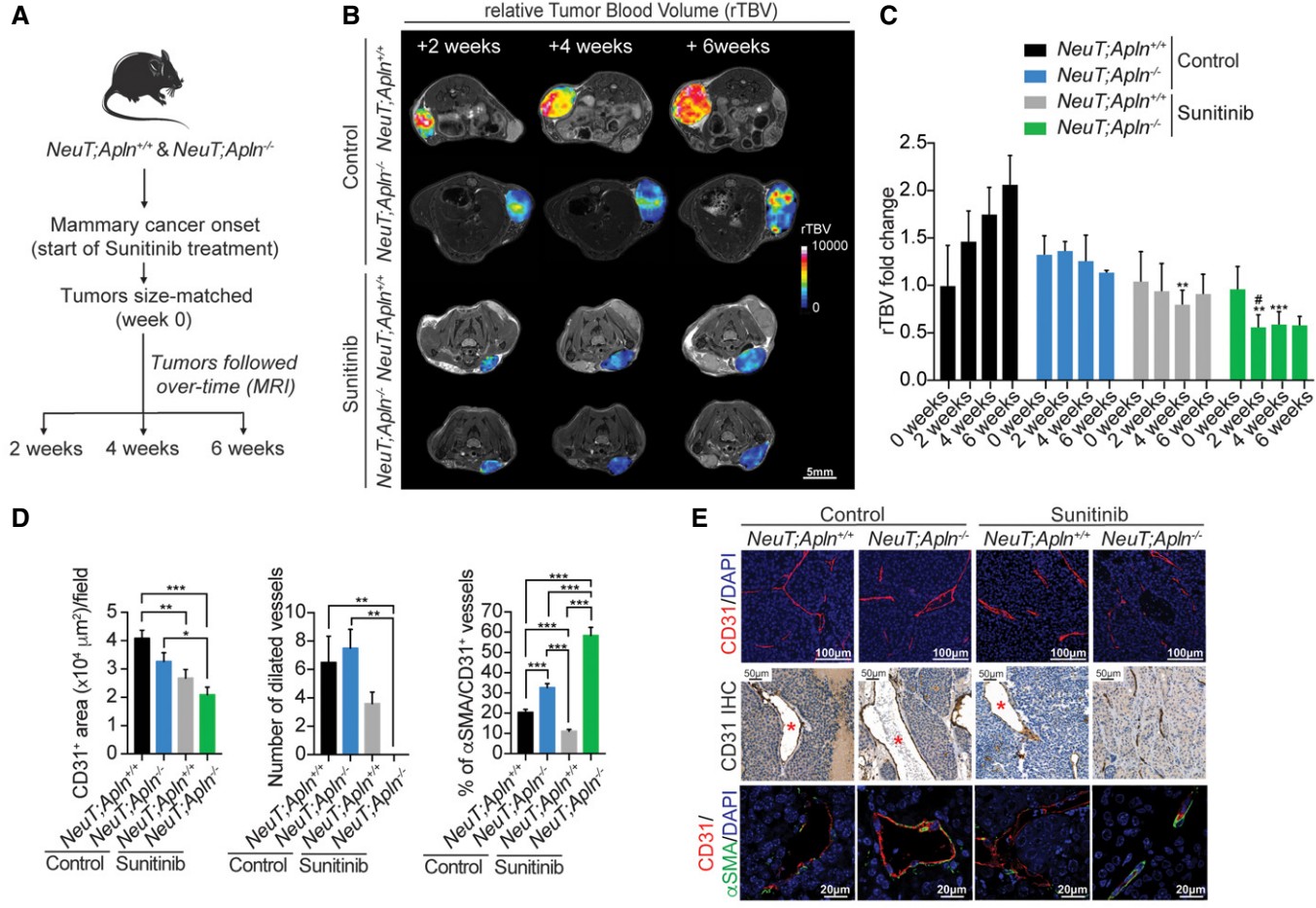

**Figure 4. Combining Apelin blockage and sunitinib treatment reduces tumor angiogenesis and normalizes tumor blood vessels.**

A   Experimental setup for the MRI (non-invasive magnetic resonance imaging) experiments.

B, C (B) Representative MRI images; scale bar = 5 mm and (C) quantification (mean ± SEM) of relative tumor blood volume (rTBV) over time after NeuT[+] mammary tumors were size-matched at 20–70 mm[3] (0 weeks). Treatments and genotypes are indicated. *NeuT;Apln*[+/+] control (n = 4), *NeuT;Apln*[−/−] control (n = 4), *NeuT; Apln*[+/+] sunitinib (n = 5), *NeuT;Apln*[−/−] sunitinib (n = 5); **P < 0.01, and ***P < 0.001 compared to untreated *NeuT;Apln*[+/+] mice and [#]P < 0.05 compared to untreated control *NeuT;Apln*[−/−] mice; two-way ANOVA. Of note, in *NeuT;Apln*[+/+] mice, only two tumors could be analyzed after 6 weeks as some mice had to be sacrificed due to the large tumor sizes following ethical guidelines. Thus, we did not perform any statistical analysis on the 6 weeks time points.

D   Analysis (mean values ± SEM) of CD31[+] area (×10[4] μm[2])/field, number of dilated tumor vessels, and percentage of alphaSMA[+] area per CD31[+] blood vessels in mammary tumors of untreated (control) and sunitinib-treated *NeuT;Apln*[+/+] and *NeuT;Apln*[−/−] mice, assessed 6 weeks after mammary tumors were size-matched. *NeuT;Apln*[+/+] control (n = 4), *NeuT;Apln*[−/−] control (n = 4), *NeuT;Apln*[+/+] sunitinib (n = 5), *NeuT;Apln*[−/−] sunitinib (n = 4); *P < 0.05; **P < 0.01; ***P < 0.001; one-way ANOVA and Kruskal–Wallis test.

E   Representative immunofluorescence and immunohistochemistry images from Fig 5D quantification. Dilated blood vessels are marked by a red asterisk. DAPI (blue) is shown as a counterstain to visualize nuclei. Scale bars = 100 μm (upper panels), 50 μm (middle panels), and 20 μm (lower panels).

blood vessel normalization as defined by less capillary leakage, reduced tissue hypoxia, and maintained pericyte coverage. While sunitinib has been tested in many clinical trials for its capability to reduce tumor growth, due to limitations in trial design and patient recruitment, a conclusive analysis whether sunitinib is increasing metastasis in human disease is still outstanding. As reported previously, sunitinib robustly induces metastases in mouse models (Ebos et al, 2009; Pàez-Ribes et al, 2009; Singh et al, 2012). Intriguingly, we did not observe an increase in metastasis upon Apelin blockage alone in both of our mammary cancer models. Importantly, Apelin blockade not only normalized blood vessels within the tumors but prevented the enhanced hypoxia and consequent metastases in animals treated with sunitinib. Thus, inhibition of distinct inducers

of angiogenesis from different receptor families provides an additional benefit in lung and mammary cancer models.

Our experimental models using E0771 mammary tumor cells, NeuT oncogene-driven spontaneous mammary cancer, as well as clonal induction of oncogenic KRas to trigger lung cancer, clearly demonstrate that loss of Apelin alone markedly reduces tumor angiogenesis, impairs tumor growth, and as a consequence improves survival of these animals. We find that impairing Apelin expression from both, the epithelial tumor cells, as well as from cells in the tumor microenvironment, is key to reduce tumor growth. We further show that Apelin depletion remodels the tumor microenvironment, by improving vessel leakiness, reducing hypoxia, and decreasing infiltration of PMN-MDSCs, while increasing NK

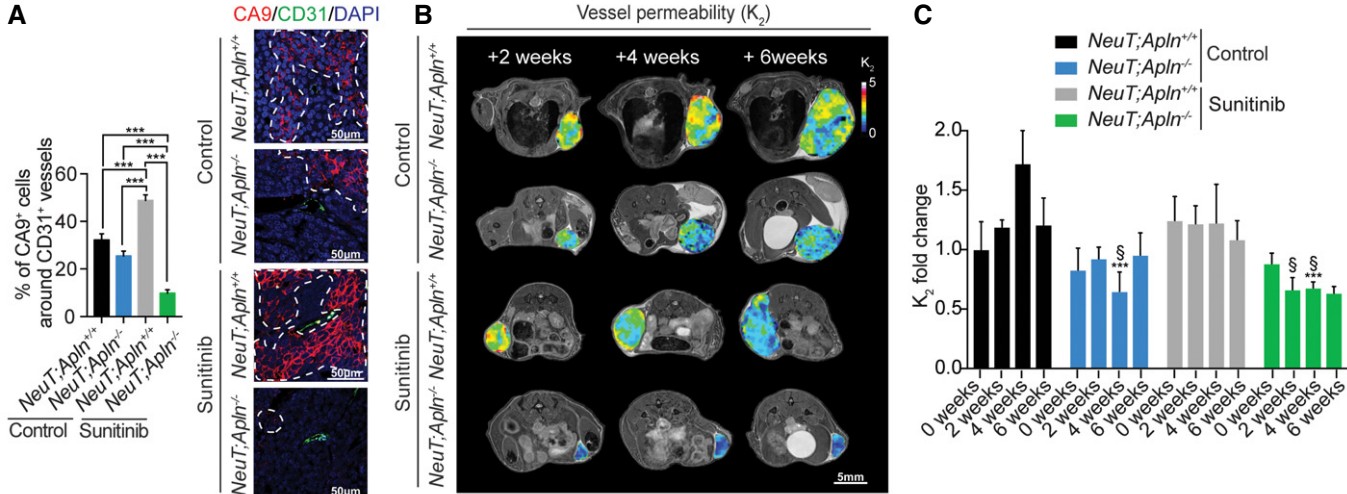

**Figure 5. Apelin inactivation reduces sunitinib therapy-induced hypoxia and vessel permeability.**

A  Percentages of CA9+ cells adjacent to CD31+ tumor blood vessels (mean ± SEM) in untreated (control) and sunitinib-treated NeuT;Apln+/+ and NeuT;Apln−/− mice, 6 weeks after mammary tumors were size-matched. NeuT;Apln+/+ control (n = 4), NeuT;Apln−/− control (n = 4), NeuT;Apln+/+ sunitinib (n = 4), NeuT;Apln−/− sunitinib (n = 4); 100–200 peri-vascular intra-tumoral regions per group were counted. ***P < 0.001; one-way ANOVA. Right panels show representative immunofluorescent images. Areas limited by dotted white lines indicate CA9+ areas. Scale bars = 50 μm.

B, C  (B) Representative MRI images; scale bar = 5 mm and (C) quantification (mean ± SEM) of vessel permeability ($K_2$) in NeuT+ mammary tumors followed over time. Treatments and genotypes are indicated. NeuT;Apln+/+ control (n = 4), NeuT;Apln−/− control (n = 5), NeuT;Apln+/+ sunitinib (n = 4), NeuT;Apln−/− sunitinib (n = 5); ***P < 0.001, compared to untreated control NeuT;Apln+/+ mice and §P < 0.05 compared to sunitinib-treated NeuT;Apln+/+ mice; two-way ANOVA. Of note, in NeuT; Apln+/+ mice, only two tumors could be analyzed after 6 weeks as some mice had to be sacrificed due to the large tumor sizes following ethical guidelines. Thus, we did not perform any statistical analysis on the 6 weeks time points.

T cells. PMN-MDSCs are an immune-suppressive subset of the neutrophil lineage that suppress immune responses in an antigen-dependent manner (Gabrilovich, 2017). MDSCs in general, can be recruited into hypoxic tissues and PMN-MDSCs in particular have been shown to be capable of promoting angiogenesis (Binsfeld et al, 2016). Importantly, MDSCs are implicated in mediating resistance to anti-angiogenic therapy by promoting new vessel growth and PMN-MDSCs numbers have been correlated with responsiveness to sunitinib therapy (Condamine et al, 2015). Our gene expression profiles further indicate that Apelin-depleted endothelial cells express factors involved in the recruitment of PMN-MDSCs. Whether the loss of Apelin affects the homing and/or local expansion of PMN-MDSCs needs to be examined in future experiments.

The effects of Apelin depletion on tumor cells and the cells of the tumor microenvironment might be model and context dependent. A recent publication found that Apelin depletion presents a double-edged sword in glioblastoma, since it enhanced tumor cell invasion, while at the same time reducing vascular density, without accompanied changes in immune cell infiltration (Mastrella et al, 2019). In contrast to this study in glioblastoma, we find a consistent benefit of depleting Apelin in epithelial breast and lung cancer, associated with normalized vessel function, decreased hypoxia, and consequently reduced metastases. Importantly, in this glioblastoma study, it was also reported that a combination of Apelin-F13A with anti-VEGFR2 therapy is superior to either intervention alone. Apelin-F13A is a poorly understood molecule proposed to be an antagonist of Apelin signaling but was shown to also exert agonistic effects (Fan et al, 2003; Medhurst et al, 2003; Lee et al, 2005). The combination of anti-angiogenic treatment with Apelin-F13A seems to be promising in glioblastoma, but due to its unclear mechanism of action its wider applicability as potential therapeutic in other types of cancers remains questionable. Nonetheless, it appears that Apelin is an ubiquitously used angiogenic pathway throughout many different types of cancers, i.e., glioblastoma, lung, and breast cancer. Whether Apelin exerts similar effects in other tumor types requires further studies.

One of the key factors limiting clinical benefits of anti-angiogenic therapy in cancer patients is to define reliable biomarkers discriminating responsive patients from non-responders. At the time of writing, to our knowledge such a reliable biomarker does not exist for guiding clinical practice. In multiple cancers, including breast and lung, Apelin expression levels in tumors inversely correlate with overall survival as well as metastasis-free survival (our data and Berta et al, 2010; Heo et al, 2012). To directly correlate Apelin expression with anti-angiogenic benefits, we accessed a cohort of renal cell carcinoma patients who received anti-angiogenic sunitinib monotherapy. High circulating Apelin levels in patients receiving anti-angiogenic therapy correlated with worse survival. Our data also show that serum Apelin levels in patients treated with sunitinib represent a potential biomarker to predict the efficacy of anti-vascular drugs and to identify patients responsive to these therapies. This is in line with recent data in 30 patients with colorectal cancer, suggesting that Apelin expression may represent a predictive biomarker for bevacizumab unresponsiveness (Zuurbier et al, 2017), independently confirming our results in human patients.

Normalization of blood vessels has become a critical concept to fully realize the clinical benefits of anti-angiogenic therapy. Although a combination therapy blocking both VEGF and

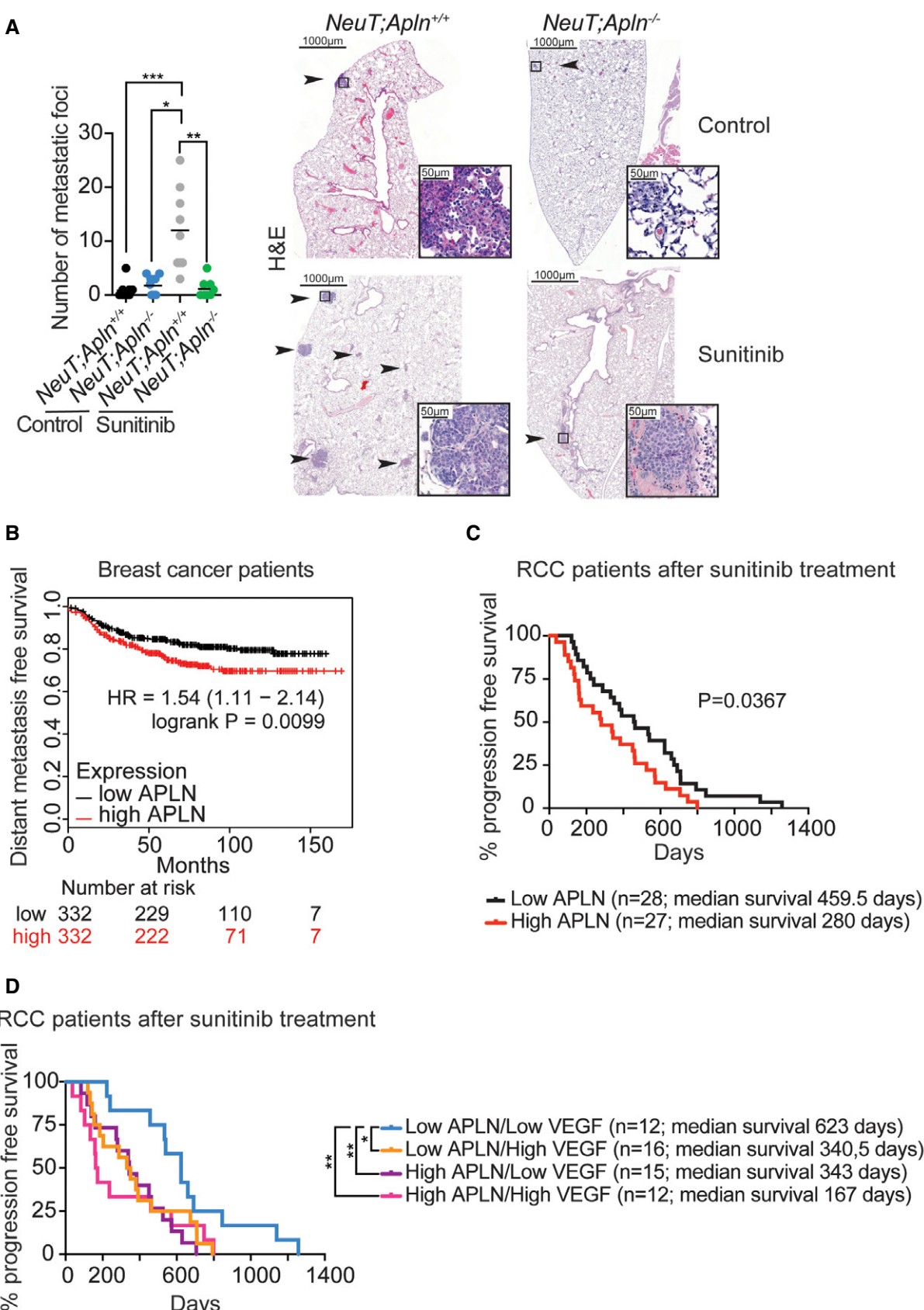

**Figure 6.**

**Figure 6. Apelin inactivation reduces sunitinib therapy-induced metastasis and stratifies sunitinib therapy responses in patients.**

A    Number of metastatic lung foci in untreated (control) and sunitinib-treated (60 mg/kg, three times a week from tumor initiation) $NeuT;Apln^{+/+}$ or $NeuT;Apln^{-/-}$ mice, assessed 6 weeks after mammary tumors were size-matched. Data of individual lung sections and means (black bars) are shown. Right panels show representative H&E images, where black arrows and insets indicate metastatic foci. Scale bars = 1,000 μm (large panels) and 50 μm (insets). *$P < 0.05$, **$P < 0.01$; ***$P < 0.001$; Kruskal–Wallis test; $n = 3$ mice per cohort and three sections per lung were analyzed.

B    Kaplan–Meier survival plot for distant metastasis-free survival from the KM-plotter database (Győrffy et al, 2013) for high and low Apelin (APLN)-expressing groups in human breast cancer. Patients were split by the median. Affymetrix Apelin ID 222856_at.

C    Kaplan–Meier plot for progression-free survival stratifying RCC patients with high and low APELIN serum levels 3–5 months after the start date of sunitinib treatment. *$P = 0.0367$; log-rank test.

D    Kaplan–Meier plots for progression-free survival in RCC patients stratified into groups of high or low levels of APELIN and VEGF. Cutoff levels were set by the median. Serum was analyzed 3–5 months after the start date of sunitinib treatment. *$P < 0.05$, **$P < 0.01$; log-rank test.

Source data are available online for this figure.

angiopoietin-2 is additive to reduce angiogenesis and tumor growth, it may promote even greater hypoxia than monotherapy (Koh *et al*, 2010; Rigamonti *et al*, 2014; Scholz *et al*, 2016). Targeting Apelin not only reduces growth and angiogenesis in our lung and breast cancer models, but also, most importantly, it does not result in increased hypoxia within the tumor microenvironment. Moreover, blocking Apelin nearly entirely prevents sunitinib-induced metastases. This is of particular importance, because drug-induced hypoxia is a key factor for drug resistance, metastasis, and metabolic reprogramming of cancer cells (Casanovas *et al*, 2005; Mazzone *et al*, 2009; Loo *et al*, 2015; Park *et al*, 2017). Preventing cancer hypoxia may therefore provide a mechanistic rationale for combination therapy with chemotherapeutics and radiation therapy, other anti-angiogenic treatments, perhaps even immune therapy (Mauceri *et al*, 1998; Jain, 2005). In summary, we have identified a druggable pathway that could stratify and protect cancer patients from resistance to current anti-angiogenic therapies and consequent metastasis, with the important feature of avoiding increased hypoxia and thus opening a new avenue for cancer treatment.

## Materials and Methods

### Mice

Apelin knockout mice ($Apln^{-/-}$ or $Apln^{-/y}$) were generated previously in our laboratories and carry a deletion of the Apelin gene in the germline (Kuba *et al*, 2007). These mice were backcrossed for at least ten generations onto a C57BL/6 background and then crossed to $LSL\text{-}KRas^{G12D}$ mice (Jackson, 2001) to generate $LSL\text{-}KRas^{G12D};$ $Apln^{-/y}$. $p53^{fl/fl}$ mice have been previously described (Jackson *et al*, 2005). MMTV-*Neu*T transgenic mice, which carry the activated c-Neu oncogene driven by a mouse mammary tumor virus (MMTV) promoter (Muller *et al*, 1988), were crossed to C57BL/6 $Apln^{-/-}$ mice to generate MMTV-*Neu*T;$Apln^{-/-}$ animals. Tumor onset of mammary tumors was determined by weekly palpation of the mammary glands; mice were sacrificed when the tumor size reached the maximum permitted under ethical guidelines. Anti-angiogenic treatment with the multimodal receptor tyrosine kinase (RTK) inhibitor sunitinib malate (1611, Biovision, 60 mg/kg body weight per dose by oral gavage, three times a week) was started only after cancer onset and followed until the mice were sacrificed. Mice were maintained in temperature-controlled conditions, and genotypes were determined by PCR analysis of genomic DNA. In all experiments, mice were randomly distributed between the different groups and only littermate mice were used as controls. All mice were maintained according to the ethical animal license protocol complying with the Austrian and European legislation.

### Histology, whole-mount sections, immunohistochemistry, and immunofluorescence analysis

Tissue samples were fixed in freshly prepared 4% paraformaldehyde (PFA) overnight at 4°C and embedded in paraffin after dehydration in ascending concentrations of ethanol. For histological analysis, 2- to 4-μm-thick paraffin sections were prepared and stained with hematoxylin and eosin. Slides were evaluated with a Zeiss Axioskop 2 MOT microscope and scanned with a *Pannoramic 250 Flash II* Scanner (3D Histech). Tumor burden was evaluated by a pathologist and also automatically scored by an algorithm programmed and executed using the *Definiens Tissue Studio* software suite. Histopathologic designations were assigned to proliferative lesions of the mammary glands in accordance with the recommendations of the Annapolis Pathology Panel (Cardiff *et al*, 2000) and the INHAND project (Rudmann *et al*, 2012). For histological analysis of lung tumors, sections from at least two different longitudinal planes of the lungs were cut and analyzed. Mitotic counts were performed manually by a pathologist; 10 representative fields were counted per tumor. The quantification of the number of dilated vessels per mammary tumor was done by a pathologist; only vessels with prominent dilated lumina (bigger than 0.25 mm in length/diameter) in intact viable tumors and inside the tumor boundaries were enumerated. Numbers of metastatic lung foci were manually counted by a pathologist; sections from three different longitudinal planes of the lungs were cut and analyzed. For immunohistochemistry, the automated Leica Bond III system was used. Ki67-positive cells and percentages of CD31 positive areas per tumor were automatically scored by an algorithm programmed and executed using the *Definiens Tissue Studio* software suite and visually controlled by a pathologist in a blinded fashion. For immunofluorescence, sections were deparaffinized in xylene and hydrated in subsequent dilutions of ethanol (100, 95, 70%), each two times for 5 min. Following antigen retrieval with unmasking solution (1:100 dilution, Vector Labs H3300) in a microwave, sections were cooled down at room temperature (RT), washed with PBS, blocked with 4% goat serum for 30 min (RT), and stained overnight with primary antibody at 4°C. The chosen fluorophore-conjugated secondary antibody was incubated for 45 min at RT, washed, and mounted in Vectashield (Vector labs, H-1000). CD31-positive areas were manually quantified; at least five representative fields were counted per tumor. To quantify the mural cell coverage of vessels, virtually all intra-tumoral $CD31^+$ vessels were manually selected and an

algorithm, programmed and executed using the *Definiens Tissue Studio* software, detected alphaSMA$^+$ (alpha smooth muscle actin) and CD31$^+$ cells. CA9$^+$ cells were quantified by manual selection of 100–200 peri-vascular intra-tumoral regions and subsequent automatic analysis by an algorithm programmed and executed using the *Definiens Tissue Studio* software. Pimonidazole positive hypoxic foci were manually counted by a pathologist. For immunohistochemistry and immunofluorescence, the primary antibodies used were as follows: anti-Ki67 (ab16667, Abcam, 1:200), anti-CD31 (DIA310, Dianova, 1:20), anti-alphaSMA (Clone 1A4, Dako, 1:20), and anti-CA9 (AF2344, R&D Systems, 1:20). For pimonidazole staining, the Hydroxyprobe™ kit (HP1-200Kit, Hydroxyprobe) was used.

## Magnetic resonance imaging (MRI)

MRI was performed on a 15.2 T Bruker system (Bruker BioSpec, Ettlingen, Germany) with a 35-mm quadrature birdcage coil. Mice were continuously monitored for tumor occurrence by palpation and MRI. When tumors reached 20–70 mm$^3$, this time point was set as imaging day 0, and then, each tumor was imaged thereafter at 2, 4, and 6 weeks. Before imaging, a tail line was inserted for delivery of contrast agent (30-gauge needle with silicon tubing). All animals were anesthetized with isoflurane (4% induction, maintenance with 1.5%). During imaging, respiration was monitored and isoflurane levels were adjusted if breathing was < 20 or > 60 breaths per minute. For measurement of tumor volume, a multi-slice multi-echo (MSME) spin-echo sequence was used [repetition time (TR)/echo time (TE) = 3,000/5.8–81.18 ms, 14 echoes, 117 μm$^2$ in-plane resolution, 0.5 mm slice thickness, number of experiments [NEX] = 1]. Tumor volume was calculated by multiplying the slice thickness with the tumor area by an investigator blinded to the treatment group.

Dynamic susceptibility contrast (DSC) perfusion MRI was collected using fast imaging with steady-state precession (FISP) with 500.6 ms temporal resolution (1 slice; TR/TE = 500/1.7 ms; flip-angle = 5 degrees; 468 × 468 μm$^2$ in-plane resolution; 1-mm slice thickness; NEX = 2; 360 repetitions) following tail vein injection of 0.05 ml of 0.25 mol/l gadolinium-based contrast agent (Magnevist, Berlex). Prior to and following DSC-MRI, a T1-weighted spin-echo dataset was acquired (0.5 mm thick slices, TR/TE 500/5.8 ms, 117 × 117 μm$^2$ resolution, 2 NEX). DSC data were processed offline using ImageJ (National Institutes of Health; rsbweb.nih.gov/ij/), and the DSCoMAN plug-in (Duke University, https://dblab.duhs.d uke.edu/wysiwyg/downloads/DSCoMAN_1.0.pdf). The analysis consisted of truncating the first five time points in the DSC-MRI time series to ensure steady-state magnetization, calculating the pre-bolus signal intensity ($S_0$) on a pixel-wise basis, converting the truncated DSC-MRI time series to a relaxivity–time curve ($\Delta R_2^*(t) = -(1/TF) \ln(S(t)/S_0)$), $S(t)$ is dynamic signal intensity curve, and correcting for the gadolinium leakage ($K_2$), as well as vessel density, as described previously (Boxerman *et al*, 2006). The effect of contrast agent leakage ($K_2$) was estimated based on pixels that exhibited signal enhancement following gadolinium injection compared to pixels that did not. Necrotic areas seen as hyperintense on T1-weighted scans were excluded as they lack viable tumor vasculature. To account for possible small differences in the amount of contrast agent administered from mouse to mouse, all tumor $K_2$ maps were subsequently corrected for $K_2$ of the muscle (muscle $K_2$ was set to 1).

## Embryoid bodies and 3D vascular sprouts

EBs were generated as previously described (Jakobsson *et al*, 2010). Briefly, mESCs were trypsinized, depleted of LIF, mixed 1:1 in case of chimeric EBs and left in low adhesion plates (MS-9096UZ, Sumitomo Bakelite Co; day 0). On day five (CCEs) or day eight (*Apln* GO and STOP mESCs), the formed EBs were transferred to a polymerized collagen I gel with addition of 30 ng/ml VEGFA164 (Peprotech; Jakobsson *et al*, 2010). The medium was changed every day after sprouting initiation. Apelin (ApelinPyr13, 1,000 nM, BACHEM H-4568) or DMSO (control) was added, when indicated, after sprouting initiation.

## Mammary cancer orthotopic model

E0771 cells were orthotopically injected in C57BL/6J mice as previously described (Ewens *et al*, 2005). In brief, cells were harvested for injection into mice by trypsin digestion for 5 min, washed in Hank's balanced salt solution, counted, diluted in this salt solution, and orthotopically injected into the fat pad of the fourth mammary gland (2.5 × 10$^5$ cells/200 μl/mouse, unless stated otherwise). Treatment with sunitinib malate (1611, Biovision, 60 mg/kg body weight per dose by oral gavage, three to five times a week), the Apelin antagonist MM54 (057-07, Phoenix Pharmaceuticals, 0.4 mg/kg body weight per dose by intra-peritoneal injection, three times a week), axitinib (A-1107, LC Laboratories, 30 mg/kg body weight per dose by oral gavage, daily), or the anti-VEGFR2 antibody (clone DC101, BioXcell, 1 mg by intra-peritoneal injection, twice a week) was started only after tumors were palpable and followed until the mice were sacrificed following the ethical guidelines. Mice were randomly distributed between the different groups. Tumors were measured using digital calipers; the size of the tumor is expressed as length (mm) × width (mm) × height (mm) = tumor size (mm$^3$).

## Sunitinib-treated RCC cancer patients

Serum samples from renal cell carcinoma (RCC) patients were collected between 2010 and 2013 3–5 months after sunitinib therapy at the Department of Urology, Semmelweis University (Budapest, Hungary). The study protocol was approved based on the ethical standards prescribed by the Helsinki Declaration of the World Medical Association and with the permission of the Scientific and Research Committee of the Hungarian Medical Research Council (ETT TUKEB 2521-0/2010-1018EKU, 153/PI/010). Samples were prepared from approximately 10 ml blood collected with BD Vacutainer Serum Separator tube (#367953). Samples were allowed to clot for 45 min at room temperature before being centrifuged at 2,650 *g* for 15 min, aliquoted, and stored at −80°C until use. Cutoff levels for high and low APELIN or VEGF levels were set by the median. Kaplan–Meier curves for progression-free survival were evaluated for all patients, and the log-rank test was used to establish the significance of the difference. Multivariate analysis of the clinical parameters was performed using the Cox regression model. Statistical analyses were performed using the PASW Statistics 18.0 package (Predictive Analytics Software, SPSS Inc., Chicago, IL, USA), the IBM SPSS Statistics 23.0 package (IBM Corp., Armonk, NY, USA), and GraphPad Prism 6.0 (GraphPad Inc., San Diego, CA, USA).

## Induction of lung cancer

Intratracheal administration of adenoviruses expressing Cre was used to specifically induce *K-Ras*$^{G12D}$ expression and *p53* deletion in pneumocytes. Administration of AdCre viruses was performed in 8- to 12-week-old mice as previously reported (DuPage *et al*, 2009). In brief, experimental animals were anesthetized with Ketasol: Xylasol and placed on a heated pad. An AdCre-CaCl$_2$ precipitate was produced by mixing 60 μl MEM, 2.5 μl AdCre ($4 \times 10^{10}$ pfu/ml; University of Iowa, Gene Transfer Vector Core Iowa, USA) and 0.6 μl CaCl$_2$ (1 M) for each mouse and incubated for 20 min at room temperature (21–22°C).

## Primary cells and cell lines

E0771 breast adenocarcinoma cells were purchased from ATCC and maintained as a monolayer in RPM1 1640 supplemented with 10% fetal bovine serum, 2 mM L-glutamine, penicillin/streptomycin (100 U Pen/ml; 0.1 mg Strep/ml), and 10 mM HEPES buffer. Primary human umbilical vein endothelial cells (HUVECs) were purchased from Life Technologies and maintained in Medium 200 (Life Technologies) supplemented with Low Serum Growth Supplement as specified by the supplier (LSGS, Life Technologies). Cells were kept at low passage and tested for mycoplasma regularly. All cells were mycoplasma negative.

## Mouse embryonic stem cells

Apelin STOP (GFP$^+$) and Apelin GO (mCherry$^+$;Cre$^+$) mouse embryonic stem cells (mESCs) as well as CCE mESCs were cultured in DMEM with 15% fetal calf serum (Invitrogen), penicillin/streptomycin (100 U Pen/ml; 0.1 mg Strep/ml), L-glutamine (2 mM), non-essential amino acids, 1 mM sodium pyruvate, 50 mM β-mercaptoethanol, and leukemia inhibitory factor (LIF). All cells were maintained at 37°C and 5% CO$_2$ conditions.

## Plasmids and retroviral infections

Apelin and Apelin receptor shRNAs were cloned into a GMPNIL retroviral vector (SSCV-GFP-miRE-PGK-Neomycin-IRES-Luciferase 2). PlatE cells were cultured at 37°C and 5% CO$_2$ in Dulbecco's modified Eagle's medium (DMEM) supplemented with 10% fetal bovine serum and transfected using the calcium phosphate co-precipitation method. Selection of E0771 was done with 400 μg/ml of neomycin. Haploid murine ESCs carrying a Tol2 gene-trap vector between exons 1 and 2 of the Apelin gene (named *Apln* STOP mESCs) were generated in-house by Haplobank. Reversion of the splice acceptor element, to genetically "repair" the mutation (named *Apln* GO), was done by infection with a retroviral plasmid encoding for Cre recombinase as well as mCherry (MSCV-mCherry-IRES-Cre).

## Vascular permeability assay

Lysinated labeled dextran (70 kDa; Invitrogen, D1818, 1.25 mg in 100 μl ddH$_2$O) was intravenously injected into the tail vein of each tumor-bearing mouse. Fifteen minutes post-injection, mice were sacrificed and tumors were harvested, followed by overnight fixation in 4% PFA. Whole-mount immunostaining of tumor samples was performed with anti-CD31 antibodies (DIA310, Dianova) followed by a goat Alexa 633 secondary antibody. Positive signals were examined by confocal microscopy and the numbers of vessels with extravasated dextran were quantified and corrected for differences in vessel densities.

## Pimonidazole hypoxia assay

Pimonidazole (Hydroxyprobe™ kit, HP1-200Kit, hydroxyprobe, 12 mg/ml in PBS) was intravenously injected at 60 mg/kg into the tail vein of each tumor-bearing mouse. Mice were sacrificed 45 min post-injection and tumors were harvested, followed by overnight fixation in 4% PFA. Immunohistochemical staining of tumor samples was performed according to the manufacturer's instructions.

## Flow cytometry

Flow cytometry sorting of endothelial cells from tumors was performed by dissociating tumors with 2 mg/ml collagenase IV (LS004186, Worthington) and 0.2 mg/ml deoxyribonuclease I (LS002138, Worthington) in RPMI medium for 45 min at 37°C. The collagenase/Dnase solution was replaced with 10 ml cold FACS buffer (PBS, 2% fetal bovine serum), and the dissociated cells were passed through a 70-μm cell strainer and then washed with 10 ml cold FACS buffer. The cells were stained with APC-conjugated anti-mouse CD31 antibody (17-0311, eBioscience, 1:100), PE-conjugated anti-mouse CD105 antibody (120408, Biolegend, 1:100), PE-Cy7-conjugated anti-mouse CD45 (103114, Biolegend, 1:400), as well as including an anti-mouse CD16/CD32 Fc block (553142, BD Biosciences, 1:100) and DAPI (D1306, Thermo Fisher Scientific, 1:500 from a 5 mg/ml stock) all diluted in FACS buffer and incubated for 20 min at 4°C. Endothelial cells were isolated by sorting for DAPI$^-$, CD45$^-$, CD31$^+$, and CD105$^+$ cells on a FACS Aria III cytometer. All data were analyzed with FlowJo v10.0.8r1.

For analysis of infiltrating immune cells, single-cell suspensions from tumors were prepared as described above. The staining was split into two panels, panel one was comprised of FITC-conjugated anti-mouse CD45 antibody (103107, eBioscience, 1:500), PerCP-Cy5.5-conjugated anti-mouse Cd11b antibody (101228, BioLegend, 1:200), PE-Cy7-conjugated anti-mouse Thy1.2 antibody (B105326, Biolegend, 1:100), BV510-conjugated anti-mouse Ly6G-antibody (127633, BioLegend, 1:100), BV711-conjugated anti-mouse NK1.1 antibody (108745, BioLegend, 1:100), BV-570-conjugated anti-mouse CD4 antibody (B100542, BioLegend, 1:250), EF450-conjugated anti-mouse CD8 antibody (48-0081-80, eBiolegend, 1:300), eFlour780-conjugated fixable viability dye (65-0865-14, eBioscience, 1:1,000) as well as including an anti-mouse CD16/CD32 Fc block (553142, BD Biosciences, 1:100). Panel 2 was comprised of AF488-conjugated anti-mouse B220/CD45R antibody (103225, BioLegend, 1:100), PerCP-Cy5.5-conjugated anti-mouse Cd11b antibody (101228, BioLegend, 1:200), PE-Cy7-conjugated anti-mouse CD45 antibody (552848, BD Biosciences, 1:500), BV510-conjugated anti-mouse Ly6G-antibody (127633, BioLegend, 1:100), BV785-conjugated anti-mouse Ly6C antibody (128041, BioLegend, 1:100), APC-conjugated anti-mouse PDCA1 antibody (17-3171-80, Thermo Fisher Scientific, 1:100), eFlour780-conjugated fixable viability dye

(65-0865-14, eBioscience, 1:1,000) as well as an anti-mouse CD16/CD32 Fc block (553142, BD Biosciences, 1:100), all diluted in FACS buffer and incubated for 30 min at 4°C. Cells were acquired on a BD LSR Fortessa. All data were analyzed with FlowJo v10.0.8r1.

Flow cytometry analysis and sorting of endothelial cells in vessel sprouts were performed as described (Jakobsson *et al*, 2010). In brief, embryoid bodies in collagen were dissociated by treatment with 2.5 mg/ml collagenase A (10103578001, Roche) in ESC media without LIF, for 45 min at 37°C. The collagenase A solution was replaced with 10 ml cold FACS buffer (PBS, 2% fetal bovine serum and 25 mM HEPES), and the EBs were passed through a 70-μm cell strainer and then washed with 10 ml cold FACS buffer. The cells were stained with APC-conjugated anti-CD31 antibodies (17-0311, eBioscience, 1:100) including an anti-mouse CD16/CD32 Fc block (553142, BD Biosciences, 1:100), all diluted in FACS buffer and incubated for 20 min at 4°C. FACS was performed using a FACS Aria III cytometer. All data were analyzed with FlowJo v10.0.8r1.

### Gene expression analysis

Total RNA of tumors, isolated mammary epithelial cells, and tumor endothelial cells were prepared using the RNeasy Mini Kit (Qiagen), according to the manufacturer's instructions. cDNA synthesis was performed using the iScript cDNA synthesis kit (Bio-Rad). RT-qPCR analyses were carried out according to the manufacturer's instructions. Values were normalized by the expression of housekeeping genes as previously described (Uribesalgo *et al*, 2011).

### Short hairpin RNA and primer sequences

The 22mer sequences are as follows:
*Apln* shRNA (MirE.2038), 5′-TAAGTGAATATCGAGCTTCTGT-3′;
*Aplnr* shRNA (MirE.2029), 5′-TTGAAAGATACAGAGCTCCTGG-3′.
Primer sequences for RT-qPCR analysis are as follows:
*Apln* forward primer: 5′-GCTCTGGCTCTCCTTGACTG-3′;
*Apln* reverse primer: 5′-CTCGAAGTTCTGGGCTTCAC-3′;
*Aplnr* forward primer 5′-GAGTTTGACTGGCCTTTTGG-3′;
*Aplnr* reverse primer 5′-GGTATCGGTCAAAGCTGAGG-3′;
*PUM1* forward primer 5′-TGTGGTCCAGAAGATGATCG-3′
*PUM1* reverse primer 5′-GGATGTGCTTGCCATAGGTG-3′;
*bActin* forward primer 5′-CGGTTCCGATGCCCTGAGGCTCTT-3′;
*bActin* reverse primer 5′-CGTCACACTTCATGATGGAATTGA-3′.

### RNA-Seq data analysis

Full RNA was isolated from sorted tumor endothelial cells (TEC) and prepared as described by Picelli *et al* (2014), with the following changes: Cells were sorted in 4 μl lysis buffer, to which oligo-dT and dNTP were added. For Tagmentation, in-house produced Tn5 was used. PolyA-mRNA was isolated from endothelial cells FACS-sorted from sprouting vessels (ES) in the presence or absence of Apelin. Both generated libraries were sequenced by 50-bp single-end Illumina mRNA sequencing. Reads were aligned using star v2.5.0a (ES) or v2.6c (TEC) in 2-pass mode, TPM estimation was performed with RSEM v1.2.25 (ES) or v1.2.28 (TEC), aligned reads were counted with HTSeq v0.6.1p1 (ES) or featurecounts subread v1.6.2 (TEC), and

differential expression analysis was performed with DESeq2 v1.10.1 (ES) or v1.18.1 (TEC). Gene sets with significant enrichment were selected based on a false discovery rate (FDR) q value cutoff of 5%. Heatmaps show scaled log2 gene expression values (log2(TPM + 1)) for differentially expressed genes (*P*adj < 0.05 and absLog2 (Foldchange) > 1, DESeq2 v1.10.1 or v1.18.1). Differentially expressed gene lists were analyzed for the enrichment of canonical pathways using IPA (ingenuity pathways analysis; Ingenuity Systems, http://www.ingenuity.com) where the effect directionality is estimated based on the provided logarithmic fold change values for the compared groups (DESeq2). GO analysis was performed with the online DAVID software.

### ELISA assays

Quantikine VEGF ELISA (DVE00, R&D Systems) and Apelin-36 ELISA (EKE-057-15, Phoenix Pharmaceuticals) kits were used. Sample preparation, standard curve generation, and measurement of samples in duplicates were performed according to the guidelines of the manufacturer.

### Human database analysis

To analyze the prognostic value of Apelin in human cancer, we performed unbiased meta-analysis of publicly available cancer microarray datasets with clinical annotation. For breast cancer, patient cohorts of 664 samples (distant metastasis-free survival) and 1,764 samples (relapse-free survival) were analyzed using the online survival analysis tool Kaplan–Meier Plotter (Győrffy *et al*, 2010; probe 222856_at); the data set GSE6532-GPL570 from the PrognoScan database was also analyzed (Mizuno *et al*, 2009). For lung cancer, a patient cohort of 1,145 samples from the Kaplan–Meier Plotter was analyzed (probe 222856_at) (Győrffy *et al*, 2013).

### Statistics

Values are given as means ± standard error of the mean (SEM) unless otherwise stated. For tumor experiments, only mice that developed tumors were included in the analysis. Mice were allocated to experimental groups on the basis of their genotype and randomized within the given group. Sample sizes were typically between $n = 3$–6 samples per group and at least $n = 7$–8 for survival curves. For animal studies, the investigator was typically blinded toward the genotype, but not the treatment group. Single comparisons were analyzed by two-tailed Student's *t*-test or Mann–Whitney test in non-normally distributed data; multiple comparisons were analyzed by one-way ANOVA, two-way ANOVA, or Kruskal–Wallis test in non-normally distributed data followed by *post hoc* tests for multiple comparisons. Normality was tested using the D'Agostino–Pearson test for $n > 7$; otherwise, Shapiro–Wilk was used. F test and Brown–Forsythe test were used to assess the equality of variances. For the Kaplan–Meier survival analysis, a log-rank test was performed. *P*-values are indicated in each figure legend. $P < 0.05$ was considered to indicate statistical significance. All exact *P*-values are listed in the Appendix Table S1. Statistical analysis was performed using GraphPad Prism (GraphPad Software, San Diego, CA, USA).

**The paper explained**

**Problem**

Depriving tumors of nutrients and oxygen by inhibiting angiogenesis (the growth of new blood vessels) using anti-angiogenic therapies is often accompanied by aberrant alteration of blood vessels and local hypoxia, which can contribute to tumor progression and increased metastasis. Existing anti-angiogenic therapies have shown limited success in the clinic.

**Results**

In this study, we show that depriving the tumor of the angiogenic molecule Apelin reduces tumor growth without increasing metastasis. Apelin inhibition blocks angiogenesis, but at the same time results in a better functionality of the remaining intra-tumoral blood vessels. In both breast and lung cancer, we found that the combination of Apelin inhibition with sunitinib, an anti-angiogenic therapy used in patients, resulted in potent reduction of tumor growth and angiogenesis. Sunitinib single therapy is often accompanied by hypoxia, tumor progression, and increased metastasis. Importantly, combining Apelin inhibition with sunitinib did not increase hypoxia and led to a reduction in metastatic burden.

**Impact**

These results identify Apelin as a druggable target for anti-angiogenic therapy in breast and lung cancer, but potentially also other tumor types. Further, our data indicate that Apelin inhibition can potentially prevent the negative consequences of current anti-angiogenic treatments and thus might provide a clinically useful target for combination therapies with other anti-angiogenic treatments.

## Data availability

The datasets produced in this study are available in the following databases:

- RNA-Seq: Gene Expression Omnibus GSE100293 (https://www.ncbi.nlm.nih.gov/geo/query/acc.cgi?acc = GSE100293)

**Expanded View** for this article is available online.

## Acknowledgements

We would like to thank G. Schmauss, T. Lendl, and K. Aumayr for expert bio-optics service as well as the VBCF Preclinical Imaging Facility, the VBCF NGS Facility and the VBCF HistoPathology Facility for their services. We also thank A. Walter and the BioImaging Austria/Correlated Multimodal Imaging Node (CMI) for their insights and expertise, P. Möseneder for his support, the IMBA/IMP Graphics Department for their assistance with the graphical abstract and the members of the Penninger laboratory for helpful discussions. I.U. is supported by an EMBO Long-term Fellowship and a Marie Curie Fellowship from the European Commission. D.H. is supported by the T. von Zastrow foundation. J.B. acknowledges funding from the Hungarian National Research, Development and Innovation Office (PD111656) and is a recipient of a Janos Bolyai Research Scholarship of the Hungarian Academy of Sciences. B.D. acknowledges funding from the Hungarian National Research, Development and Innovation Office (K109626 and KNN121510). J.M.P. is supported by grants from IMBA, the Austrian Ministry of Sciences, the Austrian Academy of Sciences, an ERC Advanced Grant, the T. von Zastrow foundation, and an Era of Hope Innovator award.

## Author contributions

Conceptualization, IU and JMP; methodology, IU, DH, YZ, AK, JL, JB, VL, MN, RK, and KK; formal analysis, IU, DH, YZ, JL, JB, RAW, VL, and MN; investigation, IU, DH, BJH, YZ, AK, JL, JB, RAW, T-PP, VL, RK, DS, LT, and LH; resources, SD, JZ, BD, MS, YC, and JMP; review and editing, IU, DH, and JMP; supervision, IU and JMP; funding acquisition, IU and JMP.

## Conflict of interest

The authors declare that they have no conflict of interest.

## For more information

(i) https://www.uniprot.org/uniprot/Q9ULZ1
(ii) http://www.informatics.jax.org/marker/MGI:1353624

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
