## [Review Process File · EMBO Molecular Medicine]

Apelin inhibition prevents resistance and metastasis associated with anti-angiogenic therapy

Iris Uribealago, David Hoffmann Yin Zhang, Anoop Kavirayani, Jelena Lazovic, Judit Berta, Maria Novatchkova, Tsung-Pin Pai, Reiner A. Wimmer, Viktória László, Daniel Schramek, Rezaul Karim, Luigi Tortola, Sumit Deswal, Lisa Haas, Johannes Zuber, Miklós Szűcs, Keiji Kuba, Balázs Dome, Yihai Cao, Bernhard J. Haubner, and Josef M. Penninger

Review timeline:

Submission date:	25 April 2018
Editorial Decision:	30 May 2018
Authors' appeal:	21 June 2018
Editorial Decision:	2 July 2018
Revision received:	17 April 2019
Editorial Decision:	13 May 2019
Revision received:	22 May 2019
Accepted:	28 May 2019

Editor: Lise Roth

Transaction Report:

1st Editorial Decision

30 May 2018

Thank you for the submission of your manuscript "Apelin inhibition prevents resistance and metastasis associated with anti-angiogenic therapy". We have now heard back from the three referees whom we asked to evaluate your manuscript.

As you will see from the enclosed reports, the referees acknowledge the potential interest of the study. However, they regret the lack of understanding of Apelin mode of action, the limited translational relevance in terms of characterization of Apln receptor antagonist or cardiotoxicity, and the unbalanced discussion in a context of controversial observations regarding anti-VEGFR treatment. Addressing all of the referees' comments would require a lot of additional work, time and effort (especially as a deeper understanding of Apelin mode of action would necessitate an endothelial cell-specific Apelin knock-out). Upon our cross-commenting exercise, the referees agreed that 3-month deadline would likely not be sufficient to provide a substantiated revision that would satisfy them. Considering the substantial points raised, and considering that we only accept papers that receive enthusiastic support upon initial review, I am afraid I see little choice but to return the manuscript to you at this point with the decision that we cannot offer to publish it.

I am sorry that I could not bring better news this time and hope that the referee comments are helpful in your continued work in this area.

***** Reviewer's comments *****

Referee #1 (Comments on Novelty/Model System for Author):

Here, the authors employ several genetic and pharmacological models, together with relevant patient liquid biopsies.

However, additional animal models are required, if the authors want to stand by their conclusions that endothelial cells are the unique targets of APLN in the tumor microenvironment (please see points 1 and 2, below).

Referee #1 (Remarks for Author):

The manuscript from UribeSalgo et al. shows that APLN inhibition impairs cancer growth via tumor angiogenesis. In addition, APLN inhibition improves sensitivity to anti-VEGFR treatments (sunitinib). Here, the authors employ several genetic and pharmacological models, together with relevant patient liquid biopsies. Although this manuscript reinforces the idea of targeting APLN for cancer therapy, the study fails in establishing the molecular mode of action of this peptide in the context of tumor growth.

Major Points

- 1- To definitively exclude non-endothelial cell target of APLN in the tumor microenvironment, beside the implanted differentiated, highly proliferative tumor cells (shAPLNR experiments), the authors should specifically knock down APLNR in host endothelial cells and monitor tumor growth. Indeed, other cells might contribute to the observed APLN action (macrophages, mesenchymal cells, cancer stem cells etc...)
- 2- In keeping with this idea, are tumor cells the privileged source of APLN in vivo? The authors should investigate APLN expression in tumor models. Is there a correlation between APLN production and tumor endothelial enrichment and angiogenic activity?
- 3- Regarding vascular normalization, the authors need to document whether endothelial proliferation & viability are affected in endothelial sprouts. In vivo, the quality of the flow in tumor vessels needs to be evaluated.
- 4- The authors propose a model in which APLNR and VEGFR pathways cooperate. Do ligands (VEGF and APLN) and drug treatments (sunitinib and MM54) act on each other receptor availability and/or activity? The authors did not report any signaling studies. Likewise, production of both VEGF and APLN should be evaluated in vitro via ELISA, along with the different treatments. Plus, from the data, it is difficult to estimate whether effects are additive or synergistic.
- 5- Previous reports suggest APLN as a druggable target in cancers, but cardiotoxicity is of high concern in Oncology. This needs to be carefully examined, notably because of the authors' ambition to combine with anti-angiogenic drugs.

Minor Points

1. Is ELABELA expression re-activated during cancer progression? Could ELABELA also contribute to APLNR angiogenic action? This is quite striking as the authors have used mESC, while ELABELA silencing limits ESC growth.
2. Was overall tumor engraftment efficiency maintained when APLNR is blocked (shRNA and MM54), similarly to the authors' observation with APLN deletion?
3. The authors should assess the endothelial behavior (sprouting/permeability) in MM54 plus VEGF conditions.
4. Could the effect of APLN deficiency on permeability due to the amount of VEGF expression?
5. The observed effect on hypoxia needs to be further characterized, with additional markers in immunohistochemistry. Notably, what kind of cells is found in these zones?
6. In Fig 3D, Ki67 images can be improved. Alternatively, BrdU might be used.
7. In Fig 5B, on MRI pictures, why the white contrast appears lower in APLN-/+ images?

Referee #2 (Remarks for Author):

In the present manuscript, the authors have examined the role of Apelin in tumor angiogenesis models.

They report that an Apelin receptor antagonist peptide and use of Apelin $-/-$ mice resulted in inhibition of tumor angiogenesis and prevention of resistance to anti-VEGFR treatment.

The study follows a series of publications implicating Apelin in physiological and pathological angiogenesis by using genetic and pharmacological tools. The authors have cited several relevant publications. The main trust of the current study is the potential role of Apelin inhibition in preventing invasiveness and metastasis following anti-VEGFR treatment in breast cancer models.

The authors have cited studies reporting that inhibition of the VEGF pathway results in promotion of metastasis, but have made no mention of the controversial nature of these observations. In this respect, I find the interpretation and discussion of the data rather one-sided and unbalanced. It would be essential to discuss and cite the relevant literature. Not only there is a lack of clinical validation, but there is considerable controversy even within preclinical models. The study by Paez-Ribes et al (Cancer Cell, 2009) has not been confirmed by other investigators in the same and in other models (for example, Singh et al J. Pathol 2012). In fact, even a study cited in the present manuscript (Rigamonti et al, Cell Rep, 2014) failed to document increased invasiveness and metastasis in the Rip-Tag model following treatment with the anti-VEGFR2 antibody DC101. What seems clear is that sunitinib and other TKIs can, in a model- and dose-dependent fashion, promote tumor cell extravasation (Chung et al, J Pathol, 2012; Welte et al, Angiogenesis, 2012). However, these are "dirty" drugs that can inhibit hundreds of kinases (see for example Kumar et al. Br J Cancer, 101: 1717-23, 2009) and attributing such effects to specific pathways is, to say the least, questionable.

In spite of such preclinical findings, there is no evidence that sunitinib treatment promotes metastasis in renal cell (Blagoev et al, Cell Rep, 3:277-81, 2013) or in breast cancer patients, even though it failed to improve PFS or OS in the latter indication (Bergh et al J. Clin Oncol.30:921-9, 2012). Failure of treatment was associated with adverse events especially in combination with chemotherapy. In this respect, the translational relevance of the findings shown in Fig. 7, showing pro-metastatic effects of sunitinib in NeuT mice, seems to say the least uncertain.

The study would benefit considerably from testing the hypothesis that Apelin inhibition improves the outcomes of anti-VEGF treatment using more clinically relevant models and more specific reagents. Several recent studies have implicated infiltration by myeloid and other pro-inflammatory cells in resistance to anti-VEGF therapy. What are the effects of Apelin inhibition in such models?

The translational relevance would be also enhanced by a more thorough characterization of the Apelin receptor antagonist. There is no mention of specificity, affinity, PK, etc.

Referee #3 (Comments on Novelty/Model System for Author):

The study lacks a mechanistic explanation for the additive effect of Apln and VEGF inhibition and the lack of hypoxia in spite of reduced vascular density in the Apln deficient condition. The technical quality of the study is OK; the tumor models are quite demanding but in-depth analyses are missing. It is quite well established that Apelin expression is elevated in a number of diseases including cancer. Whether there eventually will be therapies targeting Apelin is difficult to say.

Referee #3 (Remarks for Author):

Uribealago et al. show that ablation/inhibition of Apelin prevents VEGFR-therapy resistance and inhibition-induced metastases, and that it reduces tumor growth and angiogenesis in mouse models. Apelin levels also correlate with poor prognosis in publicly available gene-expression datasets and blood levels of Apelin correlate with poor prognoses in renal cancer patients treated with Sunitinib.

The authors have exploited a large panel of mouse tumor models:

- Breast ca models; NeuT, E0771 either in WT or Apln^{-/-} mice, alternatively with tumor cells treated with Apln shRNA or treated with the Apln inhibitor MM52
- Lung cancer models; kRasG12D in WT and Apln^{-/-} mice, p53^{f/f}; kRas; Apln^{-/y}
- Mouse tumor models {plus minus} Sunitinib
- Human cancer in the breast, lung and kidney

Data from these tumor models all convince on a correlation between Apln and VEGF expression leading to more aggressive tumor growth and shortened overall survival. The authors examine gene expression in the different conditions and find that Apln and VEGF essentially drive the same pathways. An important finding is that targeting Apln leads to reduced vascular density, without increased hypoxia. Instead vessels are normalized, leakage is reduced and metastatic propensity is significantly lowered.

This is an overall impressive study on the role of *Apln* in tumor malignancy in a wide range of mouse tumor models and in human cancer. However, the presentation is unclear and important questions remain unanswered.

Major questions/comments

1. The presentation is complex with very many models treated or not with pharmacological blockers of *Apln* or the VEGF pathway. The authors go back and forth between the different tumor models and human cancer forms, with and without treatment, making it challenging to follow the presentation even for an expert. It would have been preferable to present all the different manipulations combined for a particular cancer/model rather than continuously go between and importantly, to cut down on the many models to instead go deeper into the mechanism whereby *Apln* inhibition collaborates with VEGF inhibition to cause a more favorable tumor vasculature and reduced metastasis. Below are suggestions for how the authors would need to address the effects of *Apln* inhibition.
2. Why is the reduction in tumor vessel density in the *Apln*^{-/-} condition not accompanied by increased hypoxia? Are vessels better perfused, allowing better tissue oxygenation? Please examine the degree of vessel perfusion with and without *Apln* expression.
3. Does *Apln* suppression affect tumor inflammation, allowing anti-tumor immune responsiveness?
4. Where is *Apln* exerting its effect; in the tumor epithelium or in the endothelium? *Apln* receptors are expressed in both compartments but the relative levels are not shown.
5. Does *Apln* affect VEGF receptor expression or function?

Specific questions

6. In Fig. 1A, were E0771 shRenilla cells injected into wt C57 mice and sh*Apln* cells into *Apln*^{-/-} mice? Both cell lines need to be injected in both types of mice.
7. In Fig. 2B, exogenous *Apln* was added to the EBs destined for transcriptome analysis, moreover, very high concentrations of both VEGF and *Apln* seem to have been used - 1 microM. A suitable working concentration for at least VEGF should be 1 nM (50 ng/ml). Why was *Apln* added at all since the EBs should express endogenous *Apln* and why were such huge concentrations of the factors used?
8. Overall, this reviewer questions the usefulness of the EB model for the analysis of *Apln* transcriptional regulation and cooperatively with VEGF. It would have been much preferred to sort endothelial cells from tumors with and without *Apln* expression to test gene regulation *in vivo*. Moreover, since the proportion of EC is considerably reduced in the *Apln* sense condition (Fig. 2C), a reduction in the contribution of *Apln*^{-/-} ECs to the tip cell position is obvious. The impression from the EB data (Fig. 2A) is that *Apln* promotes EC proliferation. Did the authors test induction of EC proliferation by *Apln* on primary EC culture?
9. In Figure 5C and 6C, the statistics appear to be flawed if there were only 2 wt tumors to compare with the *Apln*/Sunitinib condition. Please remove the statistical analysis where sample size is limited to 2.
10. The authors should do an association test (e.g. Pearson chi square) to see if *Apln* levels associate with presence of metastasis in Fig. 7B.
11. In Fig. 7C, the authors conclude that high *Apln* levels correlate with poor prognosis of RCC patients treated with Sunitinib. Do high *Apln* levels also correlate with poor survival in untreated patients? This analysis is relevant if the authors believe that *Apln* affects the efficacy of Sunitinib treatment. Why don't the authors analyze the KIRC gene expression dataset from TCGA? This dataset might include untreated patients. Also, do the levels of *Apln* in blood correlate with intratumoral levels of *Apln*?

Minor questions

12. Quantifications of vessel permeability should be normalized to vessel density; it is unclear if this has been done.
13. Why is shRenilla used and not a shRNA scramble control?

Authors' appeal

21 June 2018

APPEAL LETTER

Reading the reviews might I ask you to reconsider, please. I acknowledge that the referees are very detailed, but, in some cases, it seems what they ask for is too extensive.

Referee 1: I appreciate the comments of referee 1, but one of his/her main points is asking us to redo many experiments using tissue-specific mutants. We have already done extensive animal studies for this manuscript to come to our conclusions and believe that generating endothelial specific knockout mice and using more animals at this point is not necessary. Also, in regard to other experiments he/she asks for, we have already included some data in the manuscript, e.g. the source of Apelin (e.g. please see our Suppl. Fig. 1B).

We have acquired tumor blood flow data using MRI but did not add them because we thought the functional leakiness experiment is the absolute key experiment.

As for more mechanistic data - we are prepared to analyze the immune-profile of infiltrating immune cells in the different cohorts: We already did anti-CD3 immunostaining and didn't find a difference in infiltrating CD3 cells, but we have data on changed myeloid populations, which we could add immediately. Further, we could also perform in depth immune profiling if required.

For mechanistic insights and signaling studies - we have performed signaling studies and gene expression profiling.

Here from our paper Supplementary Figure 4 legend: (E) Ingenuity Pathway Analysis (IPA) from differentially expressed genes in RNAseq analysis from CD31+ endothelial cells (ECs) isolated from sprouting EBs from either repaired *Apln* antisense cells stimulated with VEGF (30ng/mL) and Apelin (1000nM) (full presence of Apelin) or *Apln* sense sister cells stimulated with VEGF and control DMSO (total absence of Apelin) (F) Heatmap of the genes included in the VEGF IPA pathway (Figure S3F) comparing differentially expressed genes in RNAseq analysis from CD31+ endothelial cells isolated from sprouting EBs from either repaired *Apln* antisense mESCs stimulated with VEGF (30ng/mL) and Apelin (1000nM) (+*Apln*) or *Apln* sense sister cells stimulated with VEGF and control DMSO (-*Apln*). (G) Quantification of activated pathways from a phospho-kinase array in primary human umbilical venous endothelial cells (HUVECs) stimulated 5 minutes with VEGF (25ng/mL), Apelin (500nM) or both. Unstimulated HUVECs are shown as control (black bars). Data of duplicates {plus minus} S.E.M. are shown.

I do acknowledge that one could always do more, but is at the end of the day not the *in vivo* phenotype the absolute key?

Regarding the comment on the cardiotoxicity: We are aware of this issue, after all my group published the first Apelin KO mice and reported the effects on the heart (Kuba et al. *Circ. Res.* 2007). From our paper: "Apelin mutant mice are viable and fertile, appear healthy, and exhibit normal body weight, water and food intake, heart rates, and heart morphology. Intriguingly, aged Apelin knockout mice developed progressive impairment of cardiac contractility associated with systolic dysfunction in the absence of histological abnormalities. We also report that pressure overload induces upregulation of Apelin expression in the heart."

I am happy to discuss this better in the paper, and we should have done so in any case, but this was an aging effect or after overload. I also wrote the first CTLA4 knock-out paper and, based on that paper, there should be no medicine at all approved in blocking CTLA4 - these mice died after 5 weeks. Thus, blocking Apelin in cancer could clearly be of translational value.

The minor points can be readily addressed. For instance, for ELABELA, the second ligand for the Apelin receptor, we did not observe any phenotype. We can of course discuss that this second ligand should be also tested in the future for good measure.

Referee 2: this can be readily addressed. First, we concur that sunitinib can affect other tyrosine kinases. We do see, as reported by others, more metastases with sunitinib treatment in 2 different models, and these metastases are clearly reduced when we block the Apelin pathway. Thus, in our paradigms we see these effects, though controversial as correctly pointed out by the referee. We are happy to acknowledge this and can immediately edit the paper and discuss this issue to clearly point this out. I would also like to point out that we in fact also used another VEGFR blocker (Axitinib) and we also used a VEGFR blocking antibody in our E0771 breast cancer model - in both cases,

additional inhibition of the Apelin pathway significantly reduced tumor growth. This does, in our opinion, address the referee's comment on the specificity of tyrosine kinase inhibitors, like sunitinib.

We can of course provide the published data on the Apelin receptor antagonist as requested. Importantly, we do understand the general issues with antagonists, therefore we performed in vivo genetic knock-out and knock-down experiments in multiple cancer types, to underline the specificity of targeting the Apelin pathway, despite the extensive amount of work this has represented.

Referee 3: writes that "This is an overall impressive study on the role of Apln in tumor malignancy in a wide range of mouse tumor models and in human cancer".

The referee asks important questions on immune cell infiltrations and tumor perfusions/leakiness, all of which we can address and have in many parts already done so. Our apologies if that was unclear. There is also a suggestion to go deeper into the detail in one model rather than use various models – our aim was to use more models because we wanted to address how generalizable our findings were for different cancer types or whether such findings might be only relevant for one tumor type. I would like to point out these are the first tumor growth data in a genetic Apelin KO model and provide evidence for a direct role of Apelin in establishing the tumor vasculature Where Apelin and the Apelin receptor act we addressed in our E0771 model (please see Suppl. Fig. 1F).

We can address the specific question, though we disagree that the embryoid body sprouting model is not an appropriate model to study Apelin's effects on angiogenesis. The sprouting model allowed us to have a repairable Apelin mutation which is certainly not possible in vivo and therefore truly allows us to study the direct effects of Apelin depletion. In vivo one will never know what's the chicken or the egg, especially in a complex tumor setting. The other issues of statistics will of course be properly addressed.

The comments of the referees are insightful, but we think that we in fact have already addressed many of the issues. That one could always do much more, I think we can all agree, especially in terms of mechanism. However, considering the large amount of data in multiple cancer models, providing the first in vivo experimental evidence of the in vivo role of Apelin tumor biology and providing an unique new model on normalization of the tumor vasculature with the beneficial outcomes on survival and tumor growth, we would highly appreciate it if you could reconsider your decision.

At the end, as we all know, it is about the in vivo phenotypes, especially for a translational journal like EMBO Molecular Medicine, and our research aims to be translatable to the clinic, especially in cancer .

2nd Editorial Decision

2 July 2018

Thank you for your e-mail asking us to reconsider our decision on your manuscript.

I have carefully read your letter and communicated your points with one of the referees, who agreed with your line of response. In particular, the referee stated: "I agree the many tumor models are useful in particular if the presentation can be cleaned up so that gain and loss of function models are presented in a clearer manner. Importantly, the authors should provide mechanistic insights, as this was a [major] criticism" and added "mechanistic insights do not necessarily require an endothelial specific knockout model".

Therefore, after internal discussions about your manuscript with my colleagues, I would be happy to reconsider my decision and invite revision of the manuscript. Addressing the reviewers' concerns in full (apart from the apelin endothelial specific knockout) will be necessary for further considering the manuscript in our journal. EMBO Molecular Medicine encourages a single round of revision only and therefore, acceptance or rejection of the manuscript will depend on the completeness of your responses included in the next, final version of the manuscript.

Please also contact us as soon as possible if similar work is published elsewhere. If other work is published, we may not be able to extend the revision period beyond three months.

I look forward to receiving your revised manuscript.

2nd Revision - authors' response

17 April 2109

***** Point-by-point answer to the Reviewer's comments *****

Referee #1

Comments on Novelty/Model System for Author:

Here, the authors employ several genetic and pharmacological models, together with relevant patient liquid biopsies. However, additional animal models are required, if the authors want to stand by their conclusions that endothelial cells are the unique targets of APLN in the tumor microenvironment (please see points 1 and 2, below).

Remarks for Author:

The manuscript from Uribealago et al. shows that APLN inhibition impairs cancer growth via tumor angiogenesis. In addition, APLN inhibition improves sensitivity to anti-VEGFR treatments (sunitinib). Here, the authors employ several genetic and pharmacological models, together with relevant patient liquid biopsies. Although this manuscript reinforces the idea of targeting APLN for cancer therapy, the study fails in establishing the molecular mode of action of this peptide in the context of tumor growth.

Major Points:

1- To definitively exclude non-endothelial cell target of APLN in the tumor microenvironment, beside the implanted differentiated, highly proliferative tumor cells (shAPLN^R experiments), the authors should specifically knock down APLN^R in host endothelial cells and monitor tumor growth. Indeed, other cells might contribute to the observed APLN action (macrophages, mesenchymal cells, cancer stem cells etc...).

We agree with the reviewer that we cannot exclude that Apelin has non-endothelial cell targets and adding the suggested model would surely complement our data. Nevertheless, we do not claim that Apelin only targets endothelial cells but that it does have an effect on endothelial cells, which we proof in multiple experiments (please see as Figure 1C-E, 2A, S2F or S3A-B) and, importantly, that combining Apelin inhibition and an anti-angiogenic treatment can offer a significant advantage in the treatment of breast and lung cancer, which we show throughout the article using multiple models.

2- In keeping with this idea, are tumor cells the privileged source of APLN in vivo? The authors should investigate APLN expression in tumor models. Is there a correlation between APLN production and tumor endothelial enrichment and angiogenic activity?

We have performed a new set of experiments that we have included in the revised manuscript that address these questions requested by the reviewer. First, we functionally tested *in vivo* whether tumor Apelin is the only privileged source for the observed tumor growth phenotype. By injecting E0771 *shRenilla* and E0771 *shApln* mammary cancer cells into both *Apln*^{+/+} and *Apln*^{-/-} mice, we observed that Apelin produced by the tumor cells is not the only functionally

relevant source of Apelin, although it plays a relevant role together with other sources of Apelin from the microenvironment. We include these results in the new Fig.1B, which extend our previous findings and reinforce the idea that blockage of Apelin within the whole tumor would be of maximal benefit; total blockage is mimicked by our mammary and lung genetic models. Secondly, we further investigated Apelin production and tumor endothelial enrichment in different tumor models as suggested by the reviewer. By qRTPCR analysis, and using CD31 expression as indicator of vessel density, we found that, in the spontaneous NeuT-driven as well as in the orthotopic E0771 tumor models, Apelin mRNA levels positively correlate with CD31 expression and, hence, with tumor endothelial enrichment and angiogenic activity. This leads us to the conclusion that Apelin positively influences vessel growth in these tumor models, in line with our previous experiments.

Figure legend: qRTPCR of ApIn and CD31 on RNA isolated from whole mammary tumors, collected from NeuT-driven ApIn^{+/+} (n=6) and ApIn^{+/+};E0771 shRenilla (n=7). ApIn and CD31 expression was normalized to bActin and linear regression was performed to assess correlations.

3- Regarding vascular normalization, the authors need to document whether endothelial proliferation & viability are affected in endothelial sprouts.

We used endothelial sprouts because it is a well-established 3D model of angiogenesis and closely recapitulates the 3D growth requirements and properties of vessel formation. These endothelial sprouts, however, are not a feasible models to perform classical endothelial cell proliferation and viability assays. In order to be able to perform such assays, we used 2D endothelial culture models such as primary mouse endothelial cells, the mouse endothelial cell line bEnd.3 and human umbilical vein endothelial cells (HUVECs). However, in all of these models, we were unable to detect any Apelin-dependent effect on proliferation and viability using a wide range of growth conditions (full growth medium, serum deprivation, normoxia, hypoxia, different growth factor concentrations, etc), exemplified by the proliferation of HUVECs 72h after growth factor and inhibitor additions in serum deprived medium (see figure below).

Figure legend: Mean proliferation/viability of HUVECs assessed 72hrs post growth factor/inhibitor addition and normalized to 0h (n=4). 1500 HUVECs were seeded per well of a 96-well plate in full

endothelial cell growth medium (Promocell) and left to adhere overnight. The following morning the cells were switched to serum deprived medium (Medium 200 with 1% FCS) and exposed to growth factors (Apln 20ng/ml, VEGF 20 ng/ml), inhibitors (Sut = Sunitinib 200 nM, MM54 30 μ M), or combinations thereof. Proliferation/viability was assessed using the CellTiter-Glo assay.

We didn't anticipate these results, since some publications show increased Apelin-dependent proliferation in 2D culture, similar to what is observed with VEGF (e.g. Kidoya et al, 2008, EMBO J). One potential explanation for these results could be that we fail to detect meaningful Apelin receptor levels on the cell membranes of HUVECs, while the majority of Apelin receptor is localized in the cytoplasm in 2D culture conditions. Please, see also the figure below. We obtained similar results for mouse primary endothelial cells and the bEnd.3 cell line. Although this does not explain results published by others, we believe this could explain our results and, consequently, we have decided not to show any data on 2D endothelial cell cultures in our manuscript.

Figure legend: Western blot on VE-Cadherin, APLNR and GAPDH of HUVEC membrane and cytosolic fractions. The HUVECs were seeded and left to adhere overnight in full medium. The

following morning the cells were starved for 2h in medium supplemented with 1% FCS. Growth factors (20 ng/ml Apln, 20 ng/ml VEGF) were added for 2h in starvation medium. Membrane and cytosolic fractions were separated using the Qproteome Cell Compartment kit.

Despite not finding an active Apelin/ApelinR pathway in 2D endothelial cell cultures and being unable to perform the suggested experiments in our 3D model we still aimed to address this important question. Thus, we decided to conduct an RNA sequencing of sorted WT endothelial cells from tumors derived from *Apln*^{+/+} mice injected with E0771 *shRenilla* cells and compared them to Apelin-depleted endothelial cells from tumors derived from *Apln*^{-/-} mice injected with E0771 *shApln* cells. Interestingly, performing Ingenuity Pathway analysis (IPA) on the differentially regulated genes, we find that predicted decreased biological processes in Apelin-depleted tumors showed “proliferation of endothelial cells” as the top hit. This gene expression data suggests that Apelin indeed affects proliferation of endothelial cells *in vivo*.

In vivo, the quality of the flow in tumor vessels needs to be evaluated.

As the reviewer importantly suggests, we have now estimated the quality of the blood flow in the tumors by quantification of the perfusion of Dextran 2000 kDa through the tumor blood vessels. We find that blood vessels in Apelin-depleted tumors are significantly better perfused as compared to Apelin wild-type tumors (see figure below). This finding is in line with decreased leakiness in Apelin-depleted tumors and further helps to explain that, despite a reduction in vessel density, hypoxia is not increased in Apelin-depleted tumors (1C-E). This result was confirmed by additional pimonidazole staining, a chemical that covalently binds to proteins in oxygen-deprived cells. We found that Apelin-depleted tumors had less pimonidazole positive hypoxic foci (Figure1E).

Figure legend: Mean percentages (\pm SD) of Dextran 2000 kDa perfusion per tumor (n=3-4) in E0771 shRenilla or shApln mammary tumors, assessed on day 20 post-orthotopic injection into C57BL/6J *Apln*^{+/+} or *Apln*^{-/-} mice, respectively. *P<0.05; t test.

4- The authors propose a model in which APLNR and VEGFR pathways cooperate. Do ligands (VEGF and APLN) and drug treatments (sunitinib and MM54) act on each other receptor availability and/or activity? The authors did not report any signaling studies. Likewise, production of both VEGF and APLN should be evaluated *in vitro* via ELISA, along with the different treatments. Plus, from the data, it is difficult to estimate whether effects are additive or synergistic.

Consistent with previously published literature (Basagiannis, 2016, J Cell Sci), we find that VEGF negatively regulates VEGFR2 surface levels. Abolishing VEGF-induced signalling using Sunitinib increases VEGFR2 levels over the untreated control. However, we did not find an

effect of Apelin on VEGFR2 surface receptor availability (see figure below).

Figure legend: Mean percentage of VEGFR2-PE positive HUVECs over an isotype control antibody. HUVECs were seeded and left to adhere overnight in full medium. The following morning the cells were starved for 4h in medium supplemented with 1% FCS. Growth factors or inhibitors (20 ng/ml Apln, 20 ng/ml VEGF, Sunitinib (Sut) 200 nM and MM54 30 μ M) were added for 2h in starvation medium.

Further, in the membrane fractionation protocol presented in point 3 above, we failed to detect an increase in the membrane localization of the Apelin receptor in response to VEGF or Apelin addition. As outlined above, we believe that the general low Apelin receptor membrane localization could be a reason why we don't observe Apelin-induced effects in these *in vitro* assays. We further tried ELISAs for Apelin and VEGFa from cell culture supernatant of HUVECs untreated or incubated with Apelin, VEGF, MM54, Sunitinib, or MM54/Sunitinib for

24hrs, but, unfortunately, both proteins are expressed in quantities below the detection limit of commercially available ELISA assays.

5- Previous reports suggest APLN as a druggable target in cancers, but cardiotoxicity is of high concern in Oncology. This needs to be carefully examined, notably because of the authors' ambition to combine with anti-angiogenic drugs.

As we reported in Kuba, 2007, Circ Res. the Apelin knockout mice have a normal heart development with heart weights and heart-to-body ratios similar to those of WT littermate controls. Further, the left ventricular mass and its ratio to body weight were normal and the heart showed no overt structures and the expression of marker genes was normal. Thus, we concluded that complete loss of Apelin expression did not negatively influence heart development, increasing the excitement in investigating the potential targeting of Apelin for cancer treatment. We of course agree with the referee that one cannot exclude such effects in humans, which needs to be assessed in careful pre-clinical trials.

Minor Points:

1. Is ELABELA expression re-activated during cancer progression? Could ELABELA also contribute to APLNR angiogenic action? This is quite striking as the authors have used mESC, while ELABELA silencing limits ESC growth.

We have performed qRTPCR for ELABELA in several tumors from our mammary and lung mouse cancer models (NeuT-driven mammary tumors, E0771 mammary tumors and KRas-driven lung tumors). While we could detect a high level of Apelin in all of them, we were unable to detect any ELABELA transcript. This result shows that ELABELA does not appear to be re-activated during cancer progression.

2. Was overall tumor engraftment efficiency maintained when APLNR is blocked (shRNA and MM54), similarly to the authors' observation with APLN deletion?

This is an important point. Tumor engraftment efficiency was indeed maintained when the Apelin receptor is blocked in tumor cells (*shAplnR* E0771 cells compared to the control *shRenilla* E0771 cells). We also have not detected rejections of *shAplnR* E0771 cells orthotopically injected in C57B6/J mice. In addition, in the figure below we show that there is no statistical difference in the tumor sizes at day 8 after orthotopic injection in *shRenilla* versus *shAplnR* groups.

Figure legend: Mean tumor volume (\pm S.E.M.) measured at day 8 post- orthotopic injection of E0771 *shRenilla* (n=6) and *shAplnR* (n=6) cells in C57BL/6J mice. Tumor volumes were measured using a caliper. No statistical difference. t test.

3. The authors should assess the endothelial behavior (sprouting/permeability) in MM54 plus VEGF conditions.

As the reviewer suggests, we have now studied endothelial behavior in MM54 plus VEGF conditions. Wild-type mice with injected E0771 mammary cancer cells into the mammary fat pad were treated with MM54, Sunitinib or a combination of both for 17 days, subsequent to day 8 post-injection. Recapitulating the results from the genetic and orthotopic models, we find that MM54 and Sunitinib reduce vessel density in tumors, with MM54/Sunitinib showing an additive effect. Please see Figure below.

Figure legend: CD31+ area (\pm S.E.M.) of orthotopically injected E0771 *shRenilla* cells left untreated (control) or treated three times a week from day 8 after tumor injection with the Apelin antagonist alone (MM54, 0.4 μ g/g), Sunitinib alone (60mg/kg) or a combination of both. n=5 mice per cohort. Representative images are shown.

4. Could the effect of APLN deficiency on permeability be due to the amount of VEGF expression?

To answer this question, we conducted qRTPCRs and ELISAs on whole-tumor isolates from *Apln*^{+/+}; E0771 *shRenilla* and *Apln*^{-/-}; E0771 *shApln* tumors. While Apelin was strongly depleted in *Apln*^{-/-}; E0771 *shApln* tumors, we did not find a significant VEGFa deregulation neither at the mRNA nor at the protein level. Thus, it appears that Apelin does not regulate VEGF levels in the tumor microenvironment, indicating that altered VEGF expression does not readily explain how Apelin deficiency affects permeability.

Figure legend: qRTPCR of *Apln* (left) and VEGFa (middle) on whole tumor RNA-isolates from

E0771 *shRenilla* (n=8) and *shApln* (n=6) cells orthotopically injected into mammary fatpads of *Apln*^{+/+} and *Apln*^{-/-} mice, harvested day 25 post-injection. Data are shown as mean (\pm SD) relative mRNA levels normalized to *Apln*^{+/+};shRenilla. VEGFa ELISA (right) on whole tumor protein isolates from E0771 *shRenilla* (n=8) and *shApln* (n=4) cells orthotopically injected into mammary fatpads of *Apln*^{+/+} and *Apln*^{-/-} mice, harvested on day 25 post-injection. Data are shown as mean (\pm SD) VEGFa concentration per ml tumor lysate.

5. The observed effect on hypoxia needs to be further characterized, with additional markers in immunohistochemistry. Notably, what kind of cells is found in these zones?

We now further characterized hypoxia in a second model of breast cancer using the well-studied hypoxia marker pimonidazole in tumors derived from E0771 *shRenilla* vs. *shApln* cells (new Figure 1E). Confirming previous data, we observed a reduction in the number of pimonidazole positive hypoxic foci when Apelin was downregulated. This result is in accordance to our perfusion result shown above, where, by use of high-molecular Dextran, vessel perfusion was increased in tumors with targeted Apelin.

Finally, as assessed by a trained pathologist, we find that the cells in the pimonidazole positive hypoxic areas, which are typically around areas of necrosis, are predominantly mammary tumor epithelial cells. In some regions, inflammatory leukocytes (e.g. neutrophils) are present within the foci of necrosis.

6. In Fig 3D, Ki67 images can be improved. Alternatively, BrdU might be used.

We apologize if the figure was not clear, but Figure 3D was not Ki67 staining, but mitotic counts. We have now changed the previous representative images for a better H&E staining to avoid any misunderstanding. In addition, we include here quantification of Phospho-Histone H3 (PHH3) as a confirmation of the mitotic count results:

Figure legend: PHH3 positive counts (mean ± S.E.M.) and representative images of mammary tumors in untreated (control) and sunitinib-treated *NeuT;Apln^{+/+}* and *NeuT;Apln^{-/-}* mice, assessed 6 weeks after tumor onset. n=3-7; *P<0.05; one-way ANOVA

7. In Fig 5B, on MRI pictures, why the white contrast appears lower in APLN^{-/+} images?

The white contrast on the MRI pictures from Fig 5B is arbitrary, meaning that it can be modified to show better the details in the image. Of note, these images have not been used for any quantification and are solely included to illustrate the tumor features in the context of the mouse anatomy. We apologize if that was not clear in our first submission.

Referee #2

In the present manuscript, the authors have examined the role of Apelin in tumor angiogenesis models. They report that an Apelin receptor antagonist peptide and use of Apelin ^{-/-} mice resulted in inhibition of tumor angiogenesis and prevention of resistance to anti-VEGFR treatment. The study follows a series of publications implicating Apelin in physiological and pathological angiogenesis by using genetic and pharmacological tools. The authors have cited several relevant publications. The main thrust of the current study is the potential role of Apelin inhibition in preventing invasiveness and metastasis following anti-VEGFR treatment in breast cancer models.

*The authors have cited studies reporting that inhibition of the VEGF pathway results in promotion of metastasis, but have **made no mention of the controversial nature of these observations**. In this respect, I find the interpretation and discussion of the data rather one-sided and unbalanced. It*

would be essential to discuss and cite the relevant literature. Not only there is a lack of clinical validation, but there is considerable controversy even within preclinical models. **The study by Paez-Ribes et al (Cancer Cell, 2009) has not been confirmed by other investigators in the same and in other models (for example, Singh et al J. Pathol 2012).** In fact, even a study cited in the present manuscript (Rigamonti et al, Cell Rep, 2014) failed to document increased invasiveness and metastasis in the Rip-Tag model following treatment with the anti-VEGFR2 antibody DC101.

What seems clear is that sunitinib and other TKIs can, in a model- and dose-dependent fashion, promote tumor cell extravasation (Chung et al, J Pathol, 2012; Welti et al, Angiogenesis, 2012). However, these are "dirty" drugs that can inhibit hundreds of kinases (see for example Kumar et al. Br J Cancer, 101: 1717-23, 2009) and attributing such effects to specific pathways is, to say the least, questionable.

We thank the reviewer for these accurate comments. We agree with the statements in the paragraphs above regarding the differences between receptor tyrosine kinase inhibitors (TKIs) such as sunitinib, which indeed targets several different tyrosine kinases including VEGFR, and specific anti-VEGF therapy. Consequently, we have changed all relevant sections in our revised manuscript to make clearer that we use sunitinib as a current anti-angiogenic treatment rather than as a specific inhibitor of VEGFR signalling.

Further, while there is indeed controversy regarding the pro-metastatic effect of VEGF inhibition, sunitinib was consistently shown to be an inducer of invasiveness and metastasis. This was, in fact, our main motivation to explore this clinically used drug in the first place and the potential benefits that Apelin inhibition could bring in combinatorial therapy schemes. We have again changed the text in the manuscript accordingly.

In spite of such preclinical findings, there is no evidence that sunitinib treatment promotes metastasis in renal cell (Blagoev et al, Cell Rep, 3:277-81, 2013) or in breast cancer patients, even though it failed to improve PFS or OS in the latter indication (Bergh et al J. Clin Oncol.30:921-9, 2012). Failure of treatment was associated with adverse events especially in combination with chemotherapy. In this respect, the translational relevance of the findings shown in Fig. 7, showing pro-metastatic effects of sunitinib in NeuT mice, seems to say the least uncertain.

The study would benefit considerably from testing the hypothesis that Apelin inhibition improves the outcomes of anti-VEGF treatment using more clinically relevant models and more specific reagents.

We thank the reviewer for the insights and have read the cited publications with great interest. In the breast cancer trial, which tested a combination of sunitinib plus a chemotherapeutic agent against the single chemotherapeutic agent (Bergh et al J. Clin Oncol.30:921-9, 2012), patients with unresectable, locally recurrent or metastatic disease were recruited. 587 of 593 patients already presented with metastatic disease before enrolling in this clinical trial.

In our preclinical models of breast cancer, we started to administer sunitinib after tumor detection (in the spontaneous NeuT⁺ model) or tumor engraftment (in the orthotopic E0771 model) and continued to do so during the course of tumor growth. According to our hypothesis, this causes a sunitinib-dependent remodeling of the primary tumor vasculature, causing increased invasiveness and enhanced metastatic spread, which is reversed by concomitant Apelin depletion.

The patients in the clinical trial, however, present with a highly advanced and progressed disease, which might have a much more established tumor vasculature than a newly emerging breast tumor. In fact, since surgical resection is a common procedure in breast cancer, it is not even certain that these patients presented with a primary breast tumor at all before sunitinib was even administered. This would obviously obliterate any effects sunitinib might have on the vasculature of the primary tumor and could thus not increase metastatic spread. Thus, in our opinion, the way the trial was designed, especially due to the patient cohort that was recruited, precludes a conclusive interpretation whether sunitinib increases metastatic burden in primary breast cancer patients.

Similar to the breast cancer trial, the study by Blagoev et al. reports a clinical trial that recruited patients with metastatic renal cell carcinoma. They report that they do not find an acceleration in

tumor growth caused by sunitinib administration. In their discussion of the results, however, the authors themselves acknowledge some fundamental differences between animal models and the investigated clinical trial:

*“A murine model in which a small, relatively “new” tumor is being assessed **might differ from a situation** in which a patient who has tumors that are more “established” and several centimeters in size. **The former might be more dependent on “angiogenesis,” while the latter has an established vascular supply.** [...] Indeed, while sunitinib might have antiangiogenic effects in the small tumors found in mice, its activity **in humans with metastatic renal cell carcinoma might primarily be antiangiogenic but might also be more complex.** [...]”*

*We conclude that the clinical data for sunitinib do not indicate that the drug has any detrimental effect on established metastatic renal cell cancer. **We would caution that we could not draw the same conclusion for smaller, microscopic tumors, such as those that might be encountered in an adjuvant setting.** In an adjuvant setting, sunitinib is administered after a “complete” surgical resection in order to prevent or delay recurrence of occult or microscopic disease. Although there is nothing in the available evidence to suggest that adjuvant sunitinib will be harmful, in due course this question will be answered if patients enrolled in ongoing clinical trials evaluating adjuvant sunitinib experience “acceleration” of recurrences.”*

We do agree with the reasoning of the authors in the first paragraph, that metastatic disease might be less dependent on angiogenesis, blunting any sunitinib dependent effect on angiogenesis. Although our metastatic studies in mice do not include RCC, we would like to emphasize that our working model of how sunitinib increases metastatic burden in our preclinical models of mammary cancer, doesn't align with the statements made in the second paragraph about the usage of sunitinib in the adjuvant setting in the ASSURE trial, the one discussed by the authors. Sunitinib was only administered for 4 weeks after complete surgical resection of the primary tumor mass and tumor positive lymph nodes and therefore presents a different scenario to the one we test in our preclinical models (<https://clinicaltrials.gov/ct2/show/record/NCT00326898>).

To conclude, based on the current literature, we believe that the sunitinib-dependent increase in metastasis reported in preclinical studies has not been properly tested in humans and thus the hypothesis can neither be accepted nor rejected. Thus, as second-best option, we have to rely on preclinical models and these clearly show that sunitinib increases invasiveness and metastatic spread (Pàez-Ribes et al, 2009, Cancer Cell; Ebos et al, 2009, Cancer Cell; Singh et al, 2012, J Pathol). We do believe that our findings have translational relevance and hope that the presented manuscript will spark new and critical analysis of available data from clinical trials. We however agree that we need to carefully discuss our findings in relation to the literature and have now included it in our manuscript.

Several recent studies have implicated infiltration by myeloid and other pro inflammatory cells in resistance to anti-VEGF therapy. What are the effects of Apelin inhibition in such models?

We thank the referee for this great question. We have now studied the consequences of Apelin inhibition on tumor infiltrating immune cells and the results are reported in the new Figure 1F and Appendix Figure S1C. The proportion of CD45⁺ is similar between Apelin wild-type and Apelin-depleted tumors. There are also similar amounts of CD8⁺ and CD4⁺ T cells, NK cells, peripheral dendritic cells and inflammatory monocytes. However, we find a significant reduction in polymorphonuclear myeloid-derived suppressor cells (PMN-MDSCs) and a significant increase in NK T cells in Apelin-depleted vs. wild-type tumors. This is an intriguing finding, since hypoxia has been shown to cause accumulation of PMN-MDSCs, which are associated with being capable of inducing increased angiogenesis *in vivo* (Binsfeld et al, 2016, Oncotarget). We now also discuss these results in our manuscript, further characterizing the effects of Apelin targeting in cancer. This data adds important value to our findings, providing a better mechanistic understanding for the observed role of Apelin. All this data has been added now to the revised paper and is being discussed.

The translational relevance would be also enhanced by a more thorough characterization of the Apelin receptor antagonist. There is no mention of specificity, affinity, PK, etc.

We have not included this information explicitly in our manuscript as the original paper of the MM54 compound (Macaluso et al, 2011, Chem Med Chem) already characterizes it in depth. We apologize for this omission. As described in the article, which we cite in our manuscript, the MM54 compound has an affinity of $KD = 3.4 \mu M$ for the human Apelin receptor (APJ) and did not show any agonist activity in a dose-response model. This has now been also added to the revised paper.

Referee #3

Comments on Novelty/Model System for Author:

The study lacks a mechanistic explanation for the additive effect of Apln and VEGF inhibition and the lack of hypoxia in spite of reduced vascular density in the Apln deficient condition. The technical quality of the study is OK; the tumor models are quite demanding but in-depth analyses are missing. It is quite well established that Apelin expression is elevated in a number of diseases including cancer. Whether there eventually will be therapies targeting Apelin is difficult to say.

Remarks for Author:

Uribesalgo et al. show that ablation/inhibition of Apelin prevents VEGFR-therapy resistance and inhibition-induced metastases, and that it reduces tumor growth and angiogenesis in mouse models. Apelin levels also correlate with poor prognosis in publicly available gene-expression datasets and blood levels of Apelin correlate with poor prognoses in renal cancer patients treated with Sunitinib. The authors have exploited a large panel of mouse tumor models:

- Breast ca models; NeuT, E0771 either in WT or Apln^{-/-} mice, alternatively with tumor cells treated with Apln shRNA or treated with the Apln inhibitor MM52
- Lung cancer models; kRasG12D in WT and Apln^{-/-} mice, p53^{f/f}; kRas; Apln^{-/y}
- Mouse tumor models {plus minus} Sunitinib
- Human cancer in the breast, lung and kidney

Data from these tumor models all convince on a correlation between Apln and VEGF expression leading to more aggressive tumor growth and shortened overall survival. The authors examine gene expression in the different conditions and find that Apln and VEGF essentially drive the same pathways. An important finding is that targeting Apln leads to reduced vascular density, without increased hypoxia. Instead vessels are normalized, leakage is reduced and metastatic propensity is significantly lowered.

This is an overall impressive study on the role of Apln in tumor malignancy in a wide range of mouse tumor models and in human cancer. However, the presentation is unclear and important questions remain unanswered.

Major questions/comments

Major points:

1. The presentation is complex with very many models treated or not with pharmacological blockers of Apln or the VEGF pathway. The authors go back and forth between the different tumor models and human cancer forms, with and without treatment, making it challenging to follow the presentation even for an expert. It would have been preferable to present all the different manipulations combined for a particular cancer/model rather than continuously go between and importantly, to cut down on the many models to instead go deeper into the mechanism whereby Apln inhibition collaborates with VEGF inhibition to cause a more favorable tumor vasculature and reduced metastasis. Below are suggestions for how the authors would need to address the effects of Apln inhibition.

We thank this reviewer for acknowledging our efforts in using different models and apologize if the presentation has not been clear enough. We have now changed the presentation of our results to allow for an easier understanding. Specifically, we now introduce the two most important breast cancer models in main Figure 1. We moved data regarding the Apelin receptor knockdown in tumor cells to Expanded View Figure EV2B-F, since the focus of our study is on

the effect of Apelin depletion in the tumor microenvironment. Further, we moved all data regarding the lung cancer models to Expanded View Figure EV1E-H, Appendix Figure S1B and S2, to put the emphasis and focus of our study on breast cancer.

In the new paper presentation, Figure1 introduces our breast cancer models and shows how Apelin depletion remodels the tumor microenvironment.

Figure2 deepens this understanding, by showing how the transcriptome of tumor associated endothelial cells, as well as endothelial cells from 3D *in vitro* models change due to Apelin depletion.

In Figure3 we now introduce the combination of Apelin inhibition (genetic and pharmacological) and the anti-angiogenic Sunitinib treatment in breast cancer models and show that this is beneficial to survival.

We then focus on the genetic NeuT⁺ breast cancer model in Figs. 4 and 5 and establish that the combination treatment results in functional differences in blood vessel morphology (Figure4) and further that this vessel remodeling results in less hypoxia and vessel permeability in tumors (Figure5). Finally, in Figure6, we report the outcome of the combination treatment on metastatic spread/burden in mammary cancer and conclude with showing the relevance of these findings in human breast and renal cell carcinoma patients.

We moved all data on the lung cancer models, validating the results in the mammary tumor models, to the Expanded View and Appendix Figures. We sincerely hope that we have improved the presentation of the article thus making it more intuitive and easier to understand for the interested reader.

*2. Why is the reduction in tumor vessel density in the *Apln*^{-/-} condition not accompanied by increased hypoxia? Are vessels better perfused, allowing better tissue oxygenation? Please examine the degree of vessel perfusion with and without *Apln* expression.*

To answer this relevant question, we have now examined the degree of blood flow in the tumor by quantification of the perfusion of Dextran 2000 kDa through the tumor blood vessels. We find that blood vessels in Apelin-depleted tumors are significantly better perfused as compared to Apelin wild-type tumors (see Figure below). This finding is in line with decreased leakiness in Apelin-depleted tumors and further helps to explain that, despite a reduction in vessel density, hypoxia is not increased in Apelin-depleted tumors, which now we have further confirmed by additional pimonidazole staining of positive hypoxic foci (Figure1C-E). This result was confirmed by additional pimonidazole staining, a chemical that covalently binds to proteins in oxygen-deprived cells. We found that Apelin-depleted tumors had less pimonidazole positive hypoxic foci (Figure1E).

Figure legend: Mean percentages (\pm SD) of Dextran 2000 kDa perfusion per tumor (n=3-4) in E0771 *shRenilla* or *shApln* mammary tumors, assessed on day 20 post-orthotopic injection into C57BL/6J *Apln*^{+/+} or *Apln*^{-/-} mice, respectively. *P<0.05; t test.

3. Does *Apln* suppression affect tumor inflammation, allowing anti-tumor immune responsiveness?

This is a very interesting point that we have now addressed in detail. We now studied the consequences of Apelin inhibition on tumor infiltrating immune cells. The results are reported in the new Figure 1F and Appendix Figure S1C. In brief, the proportion of CD45⁺ of viable cells is similar between Apelin wild-type and Apelin-depleted tumors. There are also similar amounts of CD8⁺ and CD4⁺ T cells, NK cells, peripheral dendritic cells and inflammatory monocytes of CD45⁺ cells. However, we find a significant reduction in polymorphonuclear myeloid-derived suppressor cells (PMN-MDSC) and a significant increase in NK T cells in Apelin-depleted versus wild-type tumors. Of note, hypoxia has been shown to cause accumulation of PMN-MDSC, which can induce angiogenesis in vivo (Binsfeld et al, 2016, Oncotarget). We now also discuss these findings in our manuscript, further characterizing the effects of Apelin targeting in cancer. This data adds value to our findings, providing a potential functional insights into Apelins' role in the tumor microenvironment. All this data and a discussion of these results has been added to the revised paper.

4. Where is *Apln* exerting its effect; in the tumor epithelium or in the endothelium? *Apln* receptors are expressed in both compartments but the relative levels are not shown.

To address this question, we have now examined the relative levels of Apelin receptor (*AplnR*) in the tumor epithelium and endothelium using qRT-PCR analysis. Our new data show that isolated tumor endothelial cells express higher levels of *AplnR* compared to the tumor epithelium, presented in the new Expanded View Figure EV2C.

In addition, we have reported that *shAplnR* E0771 mammary tumors did not show any difference in growth (new Expanded View Figure EV2E) or angiogenesis (new Expanded View Figure EV2F) compared to control *shRenilla* E0771 tumors. Taken together, these results indicate that Apelin exerts its effect in the tumor microenvironment.

5. Does *Apln* affect VEGF receptor expression or function?

We have now addressed this question by investigating VEGF receptor expression in endothelial cells isolated from tumors derived of E0771 *shRenilla* and *shApln* cells injected orthotopically into *Apln*^{+/+} and *Apln*^{-/-} mice respectively (presented in Figure 2A). We find that VEGFR1,2 and 3 are not significantly deregulated in tumor associated endothelial cells when Apelin is depleted. Please see below.

Figure legend: Median abundances presented as log₂ sequencing read counts per million sequencing reads (=log₂cpm) of VEGFR1 (Flt1), VEGFR2 (Kdr) and VEGFR3 (Flt4) from a transcriptomic analysis of tumor associated endothelial cells sorted from Apelin wild-type (*Apln*^{+/+}; E0071 *shRen*) and Apelin-depleted (*Apln*^{-/-}; E0771 *shApln*) tumors. No significantly changes, DESeq2 gene expression analysis.

Specific questions

6. In Fig. 1A, were E0771 *shRenilla* cells injected into wt C57 mice and *shApln* cells into *Apln*^{-/-} mice? Both cell lines need to be injected in both types of mice.

We agree with the reviewer that this was an important control missing in our first submission. We have now added this experiment in our new Fig.1B. By injecting E0771 *shRenilla* and E0771 *shApln* mammary cancer cells into both *Apln*^{+/+} and *Apln*^{-/-} mice, we observed that Apelin produced by the tumor cells is not the only functionally relevant source of Apelin, although it plays a relevant role together with other sources of Apelin from the microenvironment. We include these results in the new Fig.1B, which extend our previous findings and reinforce the idea that blockage of Apelin within the whole tumor would be of maximal benefit; total blockage is mimicked by our mammary and lung genetic models.

7. In Fig. 2B, exogenous *Apln* was added to the EBs destined for transcriptome analysis, moreover, very high concentrations of both VEGF and *Apln* seem to have been used – 1 microM. A suitable working concentration for at least VEGF should be 1 nM (50 ng/ml). Why was *Apln* added at all since the Ebs should express endogenous *Apln* and why were such huge concentrations of the factors used?

We agree with the reviewer that a working concentration for VEGF of 1microM would be too high, and we apologize if this was not clear enough, as we do have used a concentration of 30ng/mL for the EBs and vascular organoids assays, as described in the Materials and Methods. Regarding Apelin, it has been previously used in a range from 6.5 to 652 nM in various 2D endothelial cells' assays (Kidoya, 2008, EMBO J). Since, we used Apelin in a 3D model, considering potentially limited penetration, we increased this concentration to 1 microM to make sure that we would not miss any potential effect.

The reason why we added Apelin to the EBs was to mimic the tumor scenario where, in wild-type tumors, Apelin would be produced by the endothelial cells as well as from the tumor cells. As the reviewer correctly points out, it is possible that the addition of Apelin is redundant as enough Apelin might be produced by endothelial cells to bind to all Apelin receptors from the vascular organoids, in which case the potential excess of Apelin would be innocuous. In accordance with this, the highest Apelin concentrations used in the paper cited above (Kidoya, 2008, EMBO J), were not detrimental to viability/proliferation.

8. Overall, this reviewer questions the usefulness of the EB model for the analysis of *Apln* transcriptional regulation and cooperatively with VEGF. It would have been much preferred to sort endothelial cells from tumors with and without *Apln* expression to test gene regulation *in vivo*. Moreover, since the proportion of EC is considerably reduced in the *Apln* sense condition (Fig. 2C), a reduction in the contribution of *Apln*^{-/-} ECs to the tip cell position is obvious. The impression from the EB data (Fig. 2A) is that *Apln* promotes EC proliferation. Did the authors test induction of EC proliferation by *Apln* on primary EC culture?

We thank the reviewer for this excellent comment. We have now used RNA sequencing, to obtain gene expression information from endothelial cells sorted from tumors. The data is now included into the main Figure 2A and Expanded View Figure EV3A. Interestingly, we find that Apelin depleted endothelial cells downregulate genes assigned to pathways involved in angiogenesis and immune cell recruitment. These pathways are predicted to be downregulated, resulting in decreased angiogenesis and immune cell recruitment, which is in accordance with our *in vivo* analysis of vessel density and immune cell infiltration.

We agree with the reviewer that a reduced potential for vessel formation/growth implies a reduction in the contribution to the tip cell position as well. As a consequence, and since it doesn't add new information, we have removed this result from our manuscript.

As requested, we now also assessed 2D endothelial cell cultures, specifically primary mouse endothelial cells, the mouse endothelial cell line bEnd.3 and human umbilical vein endothelial cells (HUVECs) to assess EC proliferation.

Some publications have shown a pro-proliferative effect of Apelin in these *in vitro* culture models (e.g. Kidoya et al, 2008). However, in all of these models, we were unable to detect any Apelin-dependent effect on proliferation and viability using a wide range of growth conditions (full growth medium, serum deprivation, normoxia, hypoxia, different growth factor concentrations, etc), exemplified by the proliferation of HUVECs 72h after growth factor and inhibitor additions in serum deprived medium (please see figure below).

Figure legend: Mean proliferation/viability of HUVECs assessed 72hrs post growth factor/inhibitor addition and normalized to 0h (n=4). 1500 HUVECs were seeded per well of a 96-well plate in full endothelial cell growth medium (Promocell) and left to adhere overnight. The following morning the cells were switched to serum deprived medium (Medium 200 with 1% FCS) and exposed to growth factors (Apln 20ng/ml, VEGF 20 ng/ml), inhibitors (Sut = Sunitinib 200 nM, MM54 30 μ M), or combinations thereof. Proliferation/viability was assessed using the CellTiter-Glo assay.

We believe that one potential explanation for this result could be that we fail to detect meaningful Apelin receptor levels in the cell membrane of HUVECs, with the majority of the receptor being localized to the cytoplasm (please see below). We obtained similar results for mouse primary endothelial cells and the bEnd.3 cell line.

Figure legend: Western blot on VE-Cadherin, APLNR and GAPDH of HUVEC membrane and cytosolic fractions. The HUVECs were seeded and left to adhere overnight in full medium. The following morning the cells were starved for 2h in medium supplemented with 1% FCS. Growth factors (20 ng/ml Apln, 20 ng/ml VEGF) were added for 2h in starvation medium. Membrane and cytosolic fractions were separated using the Qproteome Cell Compartment kit.

Finally to explore this question further we analysed the gene expression profile of endothelial cells from tumors derived from *Apln*^{+/+} mice injected with E0771 *shRenilla* cells and compared them to Apelin-depleted endothelial cells from tumors derived from *Apln*^{-/-} mice injected with E0771 *shApln* cells. Interestingly, performing Ingenuity Pathway analysis (IPA) on the differentially regulated genes from the RNA sequencing data in tumor endothelial cells (new Fig.2A), showed “proliferation of endothelial cells” as the top hit in the predicted decreased biological processes in Apelin-depleted tumors. This gene expression data suggests that Apelin affects proliferation of endothelial cells *in vivo*.

9. In Figure 5C and 6C, the statistics appear to be flawed if there were only 2 wt tumors to compare with the *Apln*/Sunitinib condition. Please remove the statistical analysis where sample size is limited to 2.

We apologize for this mistake and have now removed the statistical analysis from the Figures 5C and 6C (now new Figures 4C and 5C).

10. The authors should do an association test (e.g. Pearson chi square) to see if *Apln* levels associate with presence of metastasis in Fig. 7B.

When conducting a Pearson Chi Square test with the samples downloaded from kmplot.com (Gyorffy, 2013, *PLoS ONE*), we find that low Apelin expression is significantly associated with reduced presence of metastasis.

Event count	Apln Low	Apln High
Metastasis	60	87
No Metastasis	272	245

Chi-square statistic: 6.369

p-value: 0.01161

Of note, Figure 7B is now new Figure 6B.

11. In Fig. 7C, the authors conclude that high *Apln* levels correlate with poor prognosis of RCC patients treated with Sunitinib. Do high *Apln* levels also correlate with poor survival in untreated patients? This analysis is relevant if the authors believe that *Apln* affects the efficacy of Sunitinib treatment. Why don't the authors analyze the KIRC gene expression dataset from TCGA? This dataset might include untreated patients. Also, do the levels of *Apln* in blood correlate with intratumoral levels of *Apln*?

We have now extended our analysis in RCC patients and have measured the APLN levels also before sunitinib treatment as the reviewer suggests. Similar to our results in breast and lung cancer patients (Expanded View Figure EV1A and EV1E), and in accordance with our

preclinical data showing the benefit of Apelin targeting against wild-type tumors, high APLN levels indeed correlated with poor survival in RCC untreated patients, validation that high APLN levels are a bad prognosis factor in RCC patients. Therefore, these patients could benefit from blocking APLN in addition to TKRs inhibition (i.e. sunitinib treatment), as suggested by our results in mouse models where sunitinib treatment and Apelin block achieve the longest survival and slower tumor growth without increase in metastasis.

Figure legend: Kaplan-Meier plots for progression-free survival stratifying RCC patients with high and low APELIN serum levels before (left) and 3-5 months after the start date of sunitinib treatment (right). Log rank test.

We further investigated whether Apelin levels in the blood would correlate with intratumor Apelin levels, by using tumor and serum samples collected from *Apln*^{+/+}; E0771 *shRen* mice. We find a modest positive correlation between serum and tumor apelin ($R^2=0.58$, please see below).

Figure legend: Apelin levels were measured by an ELISA in serum samples and in tumor protein lysates of tumor bearing *Apln*^{+/+}; *shRenilla* mice harvested day 25 post-injection. Linear regression analysis, $R^2=0.58$

Minor questions

12. *Quantifications of vessel permeability should be normalized to vessel density; it is unclear if this has been done.*

We apologize if this information was not clear in the paper. As this reviewer correctly points out, the quantifications of vessel permeability in the paper are normalized to vessel density. Now we have explicitly included this information in the paper under the section “Materials and Methods”.

13. *Why is shRenilla used and not a shRNA scramble control?*

We agree with the reviewer that a scrambled control shRNA would also have been a valid option. However, we do believe that targeting a gene from an entirely different species is an acceptable control, and some researchers prefer using a functional shRNA as for instance stated in Moore, 2010, *Methods Mol Biol.*; Note 6

3rd Editorial Decision

13 May 2019

Thank you for the submission of your revised manuscript to EMBO Molecular Medicine. Please accept my apologies for the unusual delay in my reply, which is due to the fact that one referee did not get back to us despite several chasers. Given that the two other referees provide very similar recommendations, we prefer to make a decision now in order to avoid further delay in the process.

As you will see from the enclosed reports, both referees #1 and #3 are appreciative of the considerable improvements in the manuscript, and are now supportive of publication. I am therefore pleased to inform you that we will be able to accept your manuscript once the following final editorial amendments will be completed:

1) Referees reports:

Please address both referees' comments in writing. At this stage, we'd like you to discuss the referees' points and if you do have data at hand, we'd be happy for you to include it, however we will not ask you to provide any additional experiments. Please provide a letter including my comments, the reviewers' reports and your detailed responses to their comments (as Word file).

***** Reviewer's comments *****

Referee #1 (Remarks for Author):

The authors did a remarkable effort in this revised manuscript to give a comprehensive view of APLN involvement in angiogenesis and cancer expansion and response to treatments.

I still have few comments arising from the new experiments exposed in the rebuttal:

- Ref1, Point 3. What is the APLNR antibody used here? It is highly interesting to find that APLNR does not accumulate in membrane fractions but rather in cytosoluble ones. Can the authors check this with ImmunoFluorescence or Flow? WB is in general not the best technics to robustly detect GPCR expression, even in membrane/cyt fractions.

- Ref2, last point- regarding MM54 pharmacology, Harford-Wright et al 2017 had also characterized off-targets effects of MM54. This is also an interesting point considering the clinical potential.

Referee #3 (Comments on Novelty/Model System for Author):

The revised version of the study by Uribesalgo is considerably improved. The authors have taken the criticisms seriously and have adjusted most if not all of the points raised by the reviewers. I agree with the authors that the endothelial cell-specific knockout is not needed. The presentation is also much better now. This is a very high quality contribution spanning from mouse models to the clinic

Referee #3 (Remarks for Author):

The authors have done very important additions and modifications and thereby considerably improved their paper. I would like to congratulate the authors to very impressive work.

3rd Revision - authors' response

22 May 2019

Reply to referee's comments

Referee #1

The authors did a remarkable effort in this revised manuscript to give a comprehensive view of APLN involvement in angiogenesis and cancer expansion and response to treatments. I still have few comments arising from the new experiments exposed in the rebuttal:

- Ref1, Point 3. What is the APLNR antibody used here? It is highly interesting to find that APLNR does not accumulate in membrane fractions but rather in cytosoluble ones. Can the authors check this with ImmunoFluorescence or Flow? WB is in general not the best technics to robustly detect GPCR expression, even in membrane/cyt fractions.

To detect APLNR (also called APJ) in the Western blots, we used the Apelin Receptor Antibody 5H5L9 (ThermoFisher Scientific, #702069) at a dilution of 1:1000. Additionally, we extensively tried to detect the APLNR via FACS on HUVECs using the above-mentioned antibody and a secondary staining step, as well as the anti-human APJ APC-conjugated antibody (R&D Systems, FAB856A). On the mouse endothelial cell line bEnd.3 we tested the APJ polyclonal antibody, alexa fluor® 488 conjugated (Bioss Antibodies, bs-2430R-A488). In none of those experiments we could detect any Apelin Receptor cell surface staining over an isotype control staining. In combination with the cytosol/membrane fraction Western blots, we concluded that the Apelin Receptor, although expressed, is primarily located in the cytoplasm, but not on the cell membrane in *in vitro* cultures.

Ref. 2 last point- regarding MM54 pharmacology, Harford-Wright et al 2017 had also characterized off-targets effects of MM54. This is also an interesting point considering the clinical potential.

We agree with the reviewer that the publication of Harford-Wright et al. 2017 is a very interesting resource, considering the clinical potential of apelin inhibitors like MM54. Therefore, we have now also cited this article in the manuscript.

Referee #3

(Comments on Novelty/Model System for Author): The revised version of the study by Uribealgo is considerably improved. The authors have taken the criticisms seriously and have adjusted most if not all of the points raised by the reviewers. I agree with the authors that the endothelial cell-specific knockout is not needed. The presentation is also much better now. This is a very high quality contribution spanning from mouse models to the clinic

Referee #3 (Remarks for Author): The authors have done very important additions and modifications and thereby considerably improved their paper. I would like to congratulate the authors to very impressive work.

We thank the reviewer for these kind comments.

Corresponding Author Name: Josef M. Penninger

Manuscript Number: EMM-2018-09266-V3